# SciAgentGym: Benchmarking Multi-Step Scientific Tool-use in LLM Agents

**Yujiong Shen**[* 1] **Yajie Yang**[* 1] **Zhiheng Xi**[* 1] **Binze Hu**[1] **Huayu Sha**[1] **Qiyuan Peng**[1] **Jiazheng Zhang**[1]
**Junlin Shang**[1] **Jixuan Huang**[1] **Yutao Fan**[2] **Jingqi Tong**[1] **Ming Zhang**[1] **Shihan Dou**[1] **Zhenfei Yin**[3]
**Xingjun Ma**[1] **Lei Bai**[2] **Tao Gui**[1 †] **Qi Zhang**[1] **Xuanjing Huang**[1] **Yu-Gang Jiang**[1]

## Abstract

Scientific reasoning inherently demands integrating sophisticated toolkits to navigate domain-specific knowledge. Yet, current benchmarks largely overlook agents' ability to orchestrate tools for such rigorous workflows. To bridge this gap, we introduce **SciAgentGym**, a scalable interactive environment featuring 1,780 domain-specific tools across four natural science disciplines, supported by a robust execution infrastructure. Complementing this, we present **SciAgentBench**, a tiered evaluation suite designed to stress-test agentic capabilities from elementary actions to long-horizon workflows. Our evaluation identifies a critical bottleneck: state-of-the-art models still struggle with complex scientific tool-use, and their performance degrades substantially as interaction horizons extend. To address this, we propose **SciForge**, a data synthesis method that models the tool action space as a dependency graph to generate logic-aware training trajectories. By fine-tuning on these trajectories, our SciAgent-8B outperforms the significantly larger Qwen3-VL-235B-Instruct while exhibiting positive cross-domain transfer of scientific tool-use capabilities. These results underscore the promising potential of next-generation autonomous scientific agents.

## 1. Introduction

Modern scientific reasoning increasingly relies on tool-assisted workflows, from molecular simulations to large-scale data analysis (Wei et al., 2025; Virtanen et al., 2019). Solving these scientific problems necessitates the deployment of tools, as solutions rarely emerge from direct inference but rather through extensive trial-and-error, where

[1]Fudan University [2]Shanghai Artificial Intelligence Laboratory [3]University of Oxford. Correspondence to: Tao Gui <tgui@fudan.edu.cn>.

*Proceedings of the 43rd International Conference on Machine Learning*, Seoul, South Korea. PMLR 306, 2026. Copyright 2026 by the author(s).

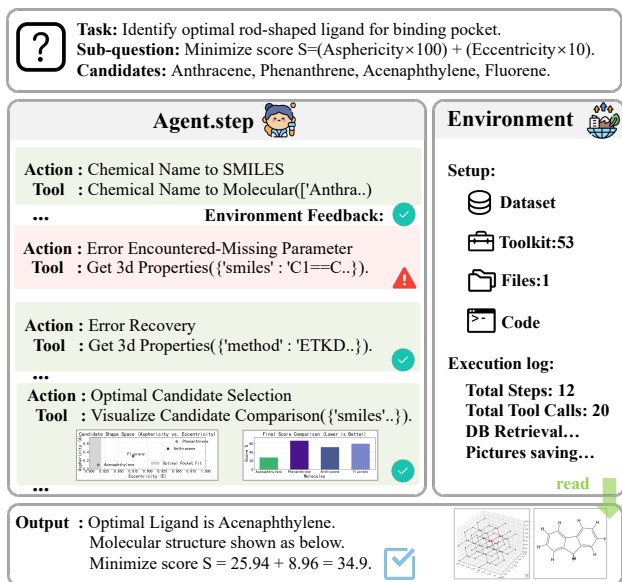

*Figure 1.* **Benchmarking multi-step scientific tool-use in Sci-AgentGym.** A representative trajectory where an LLM agent interacts with the environment to solve a complex chemistry task. This example illustrates the core agent capabilities demonstrated within our environment: orchestrating domain-specific tools, recovering from errors, and synthesizing final outputs.

LLM agents must iteratively test hypotheses and refine strategies based on execution feedback (Van et al., 2025). For instance, as illustrated in Figure 1, an agent invokes chemistry tools for ligand selection, encounters an error, and recovers before producing the final answer. This marks a paradigm shift for LLM agents from relying on internal parameterized knowledge toward reasoning through dynamic interaction and execution-based feedback.

Despite this evolving requirement, existing scientific benchmarks predominantly target static question answering and fail to capture the interactive, tool-mediated nature of actual scientific workflows (Lu et al., 2022; Wang et al., 2023). Meanwhile, general tool-use evaluations rarely reflect the breadth of domain-specific scientific tools (Xu et al., 2023; Liu et al., 2023). With the surging interest in developing capable scientific agents (Chai et al., 2025), there is an urgent need for an evaluation framework that mirrors real-

world scientific reasoning, where progress emerges through multi-turn, adaptive tool execution and iterative refinement.

To bridge this gap, we introduce **SciAgentGym** (§3), a hierarchical interactive environment designed for grounding LLM agents in multi-turn tool-use scientific reasoning tasks. Built upon an extensible architecture, the framework seamlessly integrates 1,780 domain-specific tools across Physics, Chemistry, Biology, and Materials Science, supported by essential infrastructure including a filesystem for artifact management, scientific databases for knowledge retrieval, and a python interpreter for execution. Complementing the environment is **SciAgentBench** (§4), a rigorous evaluation suite constructed to quantify the gap between tool availability and tool mastery. Spanning 259 tasks and 1,134 subquestions, the benchmark scales from elementary actions (L1) to high-fidelity, long-horizon workflows (L3). This tiered structure enables us to isolate exactly where agents struggle in complex scientific problem-solving.

Our comprehensive evaluation confirms that while tool-augmented agents outperform pure reasoning approaches, long-horizon scientific tool-use remains a distinct bottleneck. Notably, even GPT-5 achieves only a 41.3% overall success rate, with performance dropping sharply from 58.8% to 34.6% as interaction horizons increase. Fine-grained analysis reveals the root cause: weaker models frequently fall into persistent redundant tool calls, whereas stronger models manage to recover from initial errors, reflecting better adaptation under execution feedback. Current models lack a fundamental understanding of the logical dependencies between scientific tools. Without explicit structural guidance, they fail to navigate the vast tool space of potential actions.

Recognizing that standard training data lacks complex dependencies across diverse scientific tools, we design **SciForge** (§5), a data synthesis method that formalizes the tool environment from a flat collection into a dependency graph. By systematically sampling valid execution paths and synthesizing questions grounded in verified runtime traces, SciForge generates logic-aware training data. Fine-tuning on these trajectories enables our SciAgent-8B to gain +6.7%, outperforming the Qwen3-VL-235B-Instruct (Yang et al., 2025), while SciAgent-4B improves by +5.5%. These results demonstrate that scientific tool-use capabilities scale efficiently and exhibit positive cross-domain transfer.

Our key contributions are summarized as follows:

- **SciAgentGym**: An extensible environment integrating 1,780 domain-specific tools across four scientific disciplines to ground agents in multi-turn reasoning. (§3).

- **SciAgentBench**: An evaluation suite spanning elementary actions to long-horizon workflows, designed to quantify the gap between tool availability and mastery. (§4).

*Table 1.* Comparison of interactive environments and tool-use benchmarks. MM: Multimodal capabilities; Env: Stateful interactivity; DSM: Data synthesis methods; Traj: Multi-step execution trajectories.

| Works | Domain | MM | Env | DSM | Traj |
|---|---|---|---|---|---|
| *Interactive Environments* | | | | | |
| AgentBench (Liu et al., 2023) | Multi-Env | ✗ | ✓ | ✗ | ✗ |
| AgentGym (Xi et al., 2025a) | Multi-Env | ✗ | ✓ | ✗ | ✓ |
| DiscoveryWorld (Jansen et al., 2024) | Sim Science | ✗ | ✓ | ✗ | ✗ |
| MedAgentGym (Xu et al., 2025) | Biomedical | ✗ | ✓ | ✗ | ✓ |
| *Tool-Use* | | | | | |
| ToolBench (ToolLLM) (Qin et al., 2024) | General APIs | ✗ | ✗ | ✓ | ✓ |
| SciAgent (Ma et al., 2024) | Sci Reasoning | ✗ | ✗ | ✓ | ✓ |
| BFCL (Patil et al., 2025) | Func Calling | ✗ | ✗ | ✗ | ✗ |
| $\tau$-Bench (Yao et al., 2024) | Retail/Airline | ✗ | ✓ | ✓ | ✗ |
| **SciAgentGym (Ours)** | **Multi-Science** | ✓ | ✓ | ✓ | ✓ |

- **SciForge**: A graph-based data synthesis method that generates logic-aware training trajectories, enabling our 8B model to outperform 200B+ scale models. (§5).

## 2. Related Work

**LLM Agents and Interactive Environments** Interactive environments form the core infrastructure for LLM agent evaluation by enabling closed-loop perception, action, and feedback (Zhang et al., 2025; Liu et al., 2023). Existing interactive environments primarily target digital and general-purpose tasks. These range from web navigation platforms like Search-o1 (Li et al., 2025) and WebArena (Zhou et al., 2024), to code-centric environments (Yuan et al., 2025) that encapsulate interactive terminals for deterministic software engineering execution, as well as broad interaction frameworks designed to support diverse agent tasks (Xi et al., 2025a;b). In the scientific domain, DiscoveryWorld (Jansen et al., 2024) targets automated scientific discovery in a virtual environment, while MedAgentGym (Xu et al., 2025) focuses on code-centric reasoning in biomedical data science. Our SciAgentGym provides a comprehensive environment for scientific tool-use and leverages this infrastructure to construct rich, logic-aware training data.

**Scientific Reasoning Benchmarks** Scientific reasoning benchmarks have traditionally adopted a static question-answering paradigm. Datasets including ScienceQA (Lu et al., 2022), SciBench (Wang et al., 2023), MMMU (Yue et al., 2024), and GPQA (Rein et al., 2023) evaluate scientific knowledge across increasing difficulty levels. Recent benchmarks such as SuperGPQA (Du et al., 2026) further raise problem complexity while maintaining a focus on final answer accuracy. SciAgent (Ma et al., 2024) proposes tool-augmented scientific reasoning but does not provide an interactive evaluation environment. These benchmarks fail to assess iterative exploration, tool-use, or long-horizon planning. SciAgentGym instead enables feedback-driven, tool-

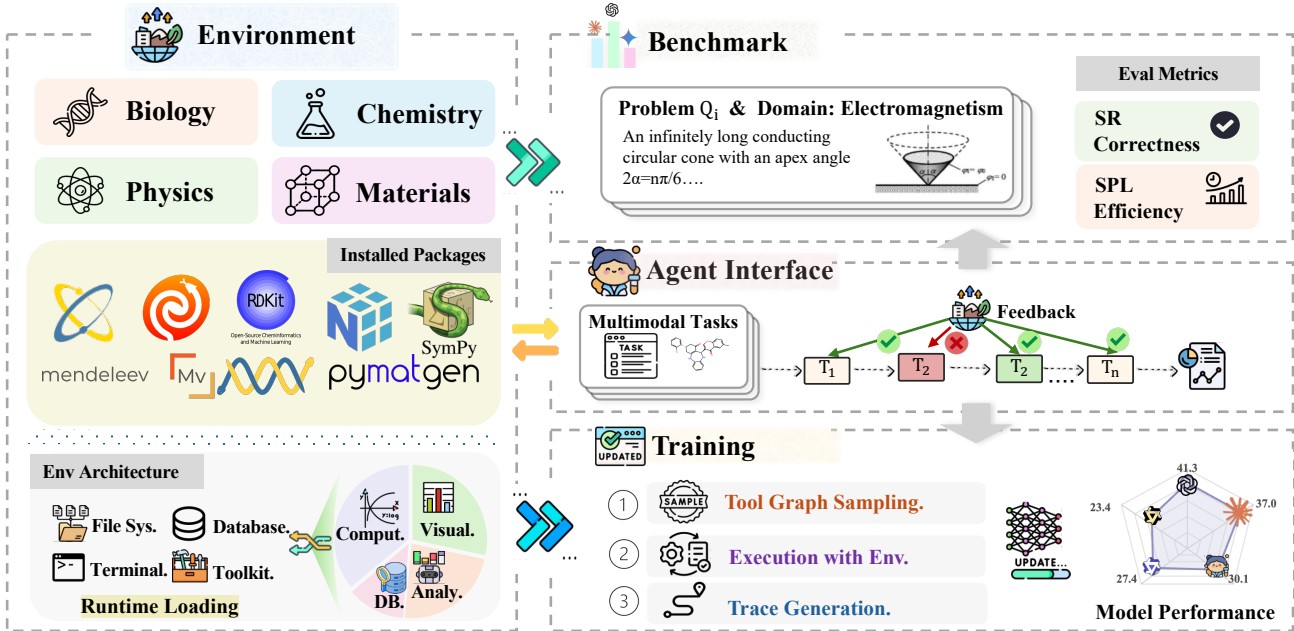

*Figure 2.* **Overview of SciAgentGym.** The left panel depicts the foundational **Environment**, integrating specialized toolkits and sandboxed infrastructure to handle multi-disciplinary multi-modal tasks. The right panel illustrates the core capabilities supported by this environment, including **SciAgentBench** for evaluation, an interactive **Agent Interface** for iterative multi-step reasoning with feedback, and a **Training** method to enhance performance via tool graph sampling and execution-grounded verification.

augmented evaluation through interactive environment, capturing agent behaviors that are absent from static paradigms. We summarize representative interactive environments and tool-use benchmarks in Table 1.

## 3. SciAgentGym

SciAgentGym provides an integrated execution environment for multi-step scientific tool-use tasks. It combines a modular architecture with a robust interaction protocol to effectively govern agent-environment dynamics.

The environment is underpinned by three core design principles: ***Type Safety***, where each tool specifies typed input/output signatures to enable automatic validation; ***Reproducibility***, ensuring all executions are recorded as structured traces with fixed random seeds; and ***Extensibility***, which organizes tools by domain via standardized protocols, enabling registration of domain custom tools.

**Environment Architecture.** SciAgentGym provides an interactive environment formalized as $\mathcal{E} = (\mathcal{S}, \mathcal{A}, \mathcal{T}, \mathcal{O})$ instantiated through four components: Toolkit, Filesystem, Databases, and Python Interpreter. The ***action space*** $\mathcal{A}$ comprises 1,780 domain-specific scientific tools, along with two infrastructure primitives: `execute_code` for Python computation and `query_database` for knowledge retrieval. The ***state*** $\mathcal{S}$ is maintained by a sandboxed filesystem, where problem assets are read-only while a writable working di-

rectory stores intermediate artifacts and execution history. The ***transition function*** $\mathcal{T}$ executes the selected action and updates the environment state. The ***observation*** $\mathcal{O}$ returns fine-grained feedback from tool execution and Python interpreter, including execution status, typed outputs, and error diagnostics. Each task runs in an isolated instance with its own registered tools and filesystem, ensuring reproducibility and avoiding cross-task contamination.

**Tool Design.** The environment comprises $|\mathcal{D}|$ scientific domains, each containing a disjoint tool set $\mathcal{V}_d$. Each tool $v \in \mathcal{V}_d$ has a signature:

$$v : (\alpha_1^v, \ldots, \alpha_{k_v}^v) \longrightarrow (\beta_1^v, \ldots, \beta_{m_v}^v), \qquad (1)$$

where $\alpha_i^v$ and $\beta_j^v$ are drawn from a scientific type system that encompasses primitive types (`Float`, `Int`), structured types (`Vector3D`, `Matrix`), and domain-specific types (`SMILES`, `ProteinStructure`).

SciAgentGym toolkit is constructed through a systematic pipeline comprising four stages: (1) analyzing scientific datasets from five source benchmarks [1] to extract discipline-specific computational patterns; (2) encapsulating established packages (`RDKit`, `ASE`, `SciPy`, `BioPython`, `PyMatGen`) into typed tools; (3) organizing tools along

---

[1] ScienceQA (Wang et al., 2023), GPQA (Rein et al., 2023), R-Bench-V (Guo et al., 2025b), BMMR (Xi et al., 2025c), SFE (Zhou et al., 2025).

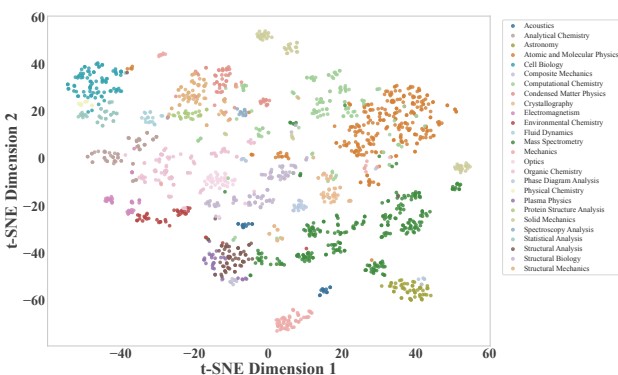

*Figure 3.* **t-SNE visualization of tool embeddings by subdomain.** Tools are colored by subdomain, illustrating the semantic diversity and broad coverage of the SciAgentGym toolkit.

two dimensions: functional categories including computation, analysis, visualization, and query, and granularity levels from atomic primitives to composite operations; and (4) automated unit testing with $\geq$75% pass-rate threshold. Figure 3 visualizes the semantic distribution. More details are provided in Appendix B.

**Closed-loop Interaction.** The environment supports three key interaction mechanisms. First, agents can query the tool registry, either loading all tools within a subdomain or selectively registering specific tools for the task. Second, upon execution failure, the environment returns fine-grained error feedback. Third, as execution proceeds, intermediate results, generated artifacts, and query records accumulate in the environment state, providing agents with an evolving reasoning context.

## 4. SciAgentBench

We introduce **SciAgentBench**, a benchmark for evaluating scientific agents on long-horizon reasoning and multi-step tool-use. The benchmark comprises 259 tasks with 1,134 sub-questions spanning four natural science domains, each verified through closed-loop interaction in SciAgentGym.

**Cross-Benchmark Standardization.** SciAgentBench constructs tasks from the same source benchmarks, unified into a single tool-use evaluation framework, enabling seamless assessment of scientific tool-use reasoning across diverse domains. To support standardized tool invocation, we extract common computational patterns from these datasets and cluster them into 1,780 validated tools with typed signatures 3. Benchmark tasks are constructed through a four-stage pipeline ensuring both difficulty and solvability: (1) aggregating approximately 5,000 candidate tasks from the source benchmarks; (2) evaluating each task using four frontier LLMs and discarding those with average accuracy exceeding 50%; (3) executing retained tasks within SciAgentGym to verify solvability, retaining only tasks that yield

*Table 2.* Statistics of SciAgentBench.

|  | Category | #Sub-questions | #Tasks | Avg. Length |
|---|---|---|---|---|
| Domain | Physics | 466 | 109 | 802 |
|  | Materials | 138 | 37 | 813 |
|  | Chemistry | 409 | 81 | 1088 |
|  | Life Sci. | 121 | 32 | 960 |
| Difficulty | L1 (Easy) | – | 54 | 507 |
|  | L2 (Medium) | – | 126 | 991 |
|  | L3 (Hard) | – | 79 | 1064 |
| **Total** |  | 1134 | 259 | 912 |

complete valid traces (recorded as golden traces); and (4) validating through domain experts that retained tasks require genuine multi-step reasoning. Details are provided in Appendix C.1.

**Dataset Statistics.** Table 2 summarizes the benchmark statistics. We design SciAgentBench to span four core scientific disciplines: Physics constitutes the largest portion with 109 tasks (42%), followed by Chemistry with 81 tasks (31%), Materials Science with 37 tasks (14%), and Life Sciences with 32 tasks (12%). The tasks are stratified by reasoning complexity: L1 ($\leq$3 steps), L2 (4–7 steps), and L3 ($\geq$8 steps), with long-horizon tasks (L2+L3) comprising 79% of the benchmark. This distribution reflects our emphasis on compositional reasoning which means each task requires agents to decompose complex scientific queries into tractable sub-problems and execute appropriate tool sequences. Furthermore, approximately 65% of tasks incorporate multimodal inputs, including molecular structure visualizations, spectral data, phase diagrams, and experimental figures, mirroring authentic scientific workflows where researchers must jointly interpret visual and textual information to reach conclusions.

**Evaluation Metrics.** Each task requires solving a sequence of sub-questions through multi-step tool invocations. At each step, the agent may either invoke a tool or perform text-based reasoning. When tools are invoked, the environment returns fine-grained feedback, either execution results or error messages. We report two complementary metrics that capture both correctness and efficiency:

- **Success Rate (SR)**: The proportion of tasks for which all sub-questions are answered correctly. Let $S_i \in \{0, 1\}$ indicate whether task $i$ is fully solved (i.e., all sub-questions are correct). Then:

$$\text{SR} = \frac{1}{N} \sum_{i=1}^{N} S_i$$

- **Success Weighted by Path Length (SPL)** (Anderson et al., 2018): A measure of path efficiency relative to an

expert-verified reference path. It is computed as:

$$\text{SPL} = \frac{1}{N} \sum_{i=1}^{N} S_i \cdot \frac{L_i}{\max(P_i, L_i)}$$

where $L_i$ is the expert-verified **reference path length** and $P_i$ is the agent's actual path length. $L_i$ serves as an efficiency baseline: when $P_i \leq L_i$, the ratio equals 1; when $P_i > L_i$, the score is discounted proportionally. Although alternative valid paths may exist, substantially longer paths ($P_i \gg L_i$) often indicate redundant tool invocations rather than meaningful alternative strategies.

# 5. SciForge: Execution-Grounded Synthesis

Beyond evaluation, we leverage SciAgentGym's executable environment and dependency modeling to synthesize high-quality training data. In this section, we introduce **SciForge**, a synthesis method that aims to address long-horizon performance degradation. By systematically constructing complex workflows validated by the environment, SciForge produces logically consistent trajectories that enable models to internalize intricate scientific invocation dependencies.

## 5.1. Tool Dependency Graph and Program Sampling

To systematically generate executable trajectories, we first construct a **Tool Dependency Graph** $\mathcal{G}_d = (\mathcal{V}_d, \mathcal{E}_d)$ for each domain $d$. This graph defines the theoretical space of composable tool sequences, where an edge $(u, v)$ signifies type-level compatibility between the output of tool $u$ and the input of tool $v$:

$$(u, v) \in \mathcal{E}_d \iff \exists i, j : \beta_i^u \preceq \alpha_j^v. \tag{2}$$

While $\mathcal{G}_d$ captures all type-level compatibility, generating concrete *executable program graphs* $\mathcal{P}$ requires a sampling process constrained by argument binding and logical stage progression. We sample from $\mathcal{G}_d$ subject to the following two constraints. The full algorithm is detailed in Appendix D.1.

First, to ensure executability, we enforce strict **Argument Binding**. For any sampled tool $v$, every input $j \in [k_v]$ must be resolved to a specific value, binding either to a type-compatible predecessor output or a root initializer:

$$\text{bind}(v, j) \in \mathcal{O}(v, j) \cup \mathcal{I}_{\tau_j^v}, \tag{3}$$

where $\mathcal{O}(v, j) = \{(u, i) : \sigma_i^u \preceq \tau_j^v\}$ denotes the set of all type-compatible predecessor outputs, and $\mathcal{I}_\tau$ denotes root initializers for type $\tau$ (domain constants, sampled parameters, or tool defaults).

Second, to simulate realistic scientific workflows (typically progressing from database query $\rightarrow$ computation $\rightarrow$ analysis $\rightarrow$ visualization), we employ a **Stage-Aware Sampling**

strategy. Instead of uniform selection, we select a predecessor $u$ for tool $v$ from candidates $\mathcal{C}$ using an $\epsilon$-greedy distribution that prioritizes stage compliance:

$$p(u \mid v) = \begin{cases} 1 - \epsilon + \frac{\epsilon}{|\mathcal{C}|} & u = u^* \\ \frac{\epsilon}{|\mathcal{C}|} & u \neq u^* \end{cases}, \tag{4}$$

where $u^*$ is the stage-compliant candidate maximizing $\mathbf{1}[\text{stage}(u) \leq \text{stage}(v)]$. This mechanism balances adherence to logical workflow order (with probability $1 - \epsilon$) against the exploration of complex, non-linear dependencies (with probability $\epsilon$) across different trajectory complexity.

## 5.2. Forward Execution with Environment

The core principle of Execution-Grounded Synthesis is that ground-truth outputs are derived from **real environment interaction**. To produce verified traces, we execute the sampled program graphs $\mathcal{P}$ by first initializing root bindings with parameters drawn from domain-specific priors (e.g., initial velocity $v_0$ or reactant concentration $C_0$). We then execute tools in topological order; at each step $i$, the environment returns a response $r_i$ (either a validated output $\mathbf{o}_i$ or fine-grained error feedback $e_i$), updating the state to $s_i$. A successful sequence yields a standard Golden Trace:

$$\mathcal{T}^* = \left[ (v_i, \mathbf{x}_i, r_i, s_i) \right]_{i=1}^{L}. \tag{5}$$

Meanwhile, we treat execution errors not as failures to be discarded, but as valuable **Error-Recovery** data. When a tool call fails with feedback $e_i$ (containing diagnostic messages), we construct corrected inputs $\mathbf{x}_i'$ for the same tool and re-execute. This process generates augmented trajectories that explicitly interleave the failed attempt with the successful correction:

$$\mathcal{T}_{\text{aug}}^* = \left[ ..., (v_i, \mathbf{x}_i, e_i, s_i), (v_i, \mathbf{x}_i', \mathbf{o}_i, s_i'), ... \right]. \tag{6}$$

By exposing the model to these "trial-and-error" sequences, $\mathcal{T}_{\text{aug}}^*$ teaches the agent both correct tool usage and adaptive error recovery strategies based on environmental feedback.

## 5.3. Trace-to-Question Generation

The final stage of our pipeline converts each verified Golden Trace $\mathcal{T}^*$ into a natural language scientific problem $Q$, yielding a fully execution-grounded dataset pair $(Q, \mathcal{T}^*)$. We employ an LLM to synthesize the problem text, guided by a domain-specific rubric $\Omega_d$ that encodes scientific laws and reasoning structures:

$$Q = \mathcal{M}_{\text{LLM}}(\mathcal{T}^*, \Omega_d). \tag{7}$$

To ensure the problem is non-trivial yet solvable, we enforce strict **Information Control** via *semantic abstraction*. While

*Table 3.* Main results on SciAgentBench. We report Success Rate (SR, %) for **without tools** and **with tools** settings. $\Delta$ denotes the improvement from tool usage. SPL measures reasoning efficiency. Best results are in **bold**; second-best are underlined.

| Model | Overall | | | | By Subject (w/ Tools) | | | | By Difficulty (w/ Tools) | | |
|---|---|---|---|---|---|---|---|---|---|---|---|
| | w/o Tools | w/ Tools | $\Delta$ | SPL | Phys. | Chem. | Mat. | Life | L1 | L2 | L3 |
| *Closed-Source Models* | | | | | | | | | | | |
| GPT-5 | **32.3** | **41.3** | +9.0 | 0.24 | 46.3 | **43.8** | 28.6 | **32.3** | **58.8** | 38.4 | **34.6** |
| Grok-4-1 | 30.4 | 40.3 | +9.9 | 0.25 | **47.2** | 38.2 | **32.4** | 30.0 | 50.0 | **43.9** | 28.6 |
| Claude-Sonnet-4 | 22.4 | 35.9 | **+13.5** | 0.19 | 39.4 | 39.5 | 27.0 | 25.0 | 57.4 | 36.5 | 20.3 |
| Gemini-2.5-Flash | 28.5 | 32.7 | +4.2 | 0.21 | 38.3 | 32.4 | 28.6 | 17.2 | 48.0 | 33.1 | 22.1 |
| Gemini-2.5-Pro | 24.8 | 32.6 | +7.8 | 0.21 | 37.3 | 35.1 | 26.5 | 18.8 | 54.2 | 33.6 | 17.3 |
| O3 | 26.6 | 32.0 | +5.4 | **0.26** | 35.5 | 37.3 | **32.4** | 6.5 | 47.9 | 31.4 | 23.1 |
| O4-mini | 27.8 | 31.1 | +3.3 | 0.24 | 31.2 | 35.5 | 30.6 | 20.0 | 51.0 | 35.0 | 11.7 |
| Gemini-2.5-Pro-Think | 28.9 | 28.8 | -0.1 | 0.19 | 33.3 | 28.9 | 21.2 | 21.9 | 51.0 | 28.8 | 14.5 |
| GPT-4o | 17.1 | 18.7 | +1.6 | 0.14 | 21.3 | 20.5 | 8.6 | 16.0 | 36.0 | 17.4 | 9.2 |
| *Open-Source Large Models (>30B)* | | | | | | | | | | | |
| GLM-4.6V | 26.0 | 30.9 | +4.9 | 0.25 | 30.9 | 37.5 | 22.2 | 18.8 | 48.8 | 27.5 | 22.2 |
| Qwen3-VL-235B-Think | 24.4 | 28.0 | +3.6 | 0.16 | 30.6 | 29.5 | 22.9 | 22.6 | 46.8 | 27.4 | 17.4 |
| Qwen3-VL-235B-Inst | 23.0 | 23.9 | +0.9 | 0.16 | 28.1 | 26.5 | 5.0 | 17.2 | 57.1 | 22.6 | 4.6 |
| Qwen3-VL-32B-Think | 24.4 | 27.9 | +3.5 | 0.17 | 33.0 | 31.2 | 8.8 | 22.6 | 45.3 | 27.3 | 16.9 |
| Qwen3-VL-32B-Inst | 22.8 | 27.4 | +4.6 | 0.15 | 31.8 | 29.3 | 20.0 | 16.1 | 47.1 | 27.3 | 14.5 |
| *Open-Source Small & Medium Models (≤30B)* | | | | | | | | | | | |
| Qwen3-VL-8B-Inst | 18.4 | 23.4 | +5.0 | 0.09 | 24.0 | 28.6 | 7.1 | 24.1 | 44.4 | 25.2 | 7.0 |
| SciAgent-8B | 23.3 +4.9 | 30.1 +6.7 | +6.8 | 0.16 | 33.0 +9.0 | 35.2 +6.6 | 9.1 +2.0 | 31.0 +6.9 | 45.5 +1.1 | 30.5 +5.3 | 20.3 +13.3 |
| Qwen3-VL-4B-Inst | 17.0 | 19.7 | +2.7 | 0.10 | 23.8 | 20.6 | 10.3 | 13.3 | 48.8 | 16.5 | 8.3 |
| SciAgent-4B | 17.4 +0.4 | 25.2 +5.5 | +7.8 | 0.13 | 28.4 +4.6 | 28.4 +7.8 | 14.7 +4.4 | 19.4 +6.1 | 46.5 -2.3 | 24.3 +7.8 | 14.5 +6.2 |
| Pixtral-12B | 7.8 | 7.2 | -0.6 | 0.07 | 7.5 | 6.3 | 5.9 | 10.0 | 16.0 | 5.0 | 5.1 |
| *Average* | 23.3 | 28.3 | +4.9 | 0.18 | 31.6 | 30.8 | 19.0 | 20.1 | 47.4 | 28.0 | 16.4 |

the final answer is requested, precise intermediate execution outputs $\mathbf{o}_1, ..., \mathbf{o}_{L-1}$ are concealed from the problem text. Instead, quantitative values are mapped to qualitative descriptors (e.g., "turbidity = 47.3" $\rightarrow$ "slightly cloudy"), providing necessary reasoning context without leaking exact intermediate solutions.

# 6. Experiments

## 6.1. Experimental Setup

**Models.** We evaluate a set of recent *multimodal* large language models that support *tool calling* for agentic problem solving, spanning both proprietary and open-source families: OpenAI GPT series (GPT-4o (OpenAI, 2023), GPT-5 (Singh et al., 2025)), Anthropic Claude series (Claude-4-Sonnet (Anthropic, 2025)), Google Gemini series (Gemini-2.5-Flash, Gemini-2.5-Pro) by Team et al. (2023), Qwen3-VL series (Qwen3-VL-4B-Inst, Qwen3-VL-8B-Inst, Qwen3-VL-32B-Inst, Qwen3-VL-235B-Inst, and their Thinking variants) by Yang et al. (2025), and GLM-4.6v (Zeng et al., 2025), etc.

**Testing Paradigm.** We evaluate agents under two settings: *with tools* and *without tools*. In the *with-tools* setting, agents interact with tools via a ReAct-style (Yao et al., 2022) loop, with tool interfaces adapted to each model's native function-calling format (e.g., OpenAI tool schemas, GLM's tool

format). In the *without-tools* setting, agents solve tasks using chain-of-thought prompting alone, serving as a baseline to isolate the contribution of tool-use. Detailed inference settings and prompt templates are provided in Appendix E.1.

**Training hyperparameters.** Using our execution-verified trajectories, we fine-tune Qwen3-VL-8B and Qwen3-VL-4B with SFT at different data scales. In the main results table, we report the best-performing checkpoints: *SciAgent-8B* and *SciAgent-4B* trained on 11,074 trajectories. Full training settings and additional runs are provided in Appendix E.2 .

## 6.2. RQ1: Can current models handle long-horizon, multi-modal scientific tasks?

Table 3 presents evaluation results on SciAgentBench. Our primary goal is to examine how models leverage scientific tools when confronted with complex problems that cannot be solved through parametric knowledge or simple reasoning alone. Based on the experimental results, we draw the following key findings:

**Current models struggle with long-horizon, multi-modal scientific tasks.** Performance degrades substantially as task complexity increases. Across all models, accuracy drops sharply from L1 to L3 difficulty levels. For instance, GPT-5 declines from 58.8% (L1) to 34.6% (L3), and Claude-Sonnet-4 drops from 57.4 % to just 20.3%. On average,

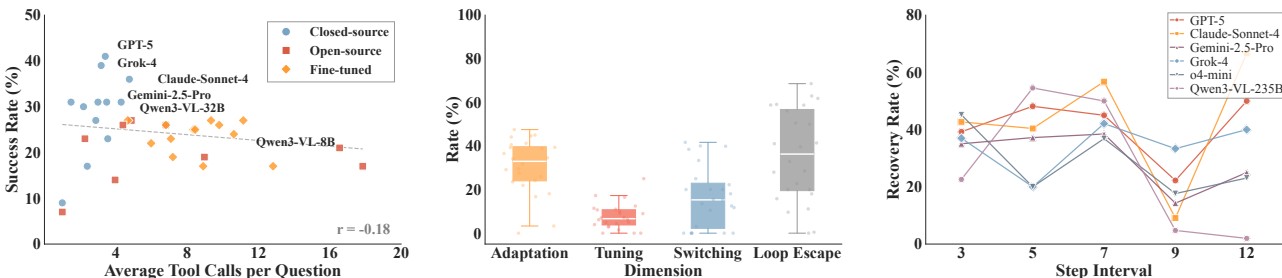

*Figure 4.* **Failure analysis in SciAgentBench. (Left)** Tool-call frequency vs. Success Rate; the negative correlation ($r = -0.18$) implies ineffective loops. **(Middle)** Feedback metrics: Adaptation (responding to errors), Tuning (parameter refinement), Switching (strategy pivoting), and Loop Escape (1- identical repetition rate). **(Right)** Error recovery rates over step intervals.

models achieve 47.4% on L1 tasks but only 16.4% on L3 tasks, representing a 65.4% relative performance degradation. This trend holds consistently across both closed-source and open-source models, indicating that current models face fundamental challenges in maintaining reasoning coherence over extended multi-step scientific workflows.

**Tool Augmentation Remains Essential for Scientific Problem Solving.** Our results underscore that tool-augmented reasoning is not merely beneficial but essential for tackling complex scientific problems. For standard instruction-following models, the interactive ReAct paradigm yields substantial improvements: Claude-Sonnet-4 improves from 22.4% to 35.9% with tools, while SciAgent-8B gains +6.8%. Similarly, models equipped with reinforced chain-of-thought capabilities (Thinking) benefit from tool integration: Qwen3-VL-235B-Think and Qwen3-VL-32B-Think achieve gains of +3.6% and +3.5% respectively. These findings suggest that pure reasoning cannot fully substitute for the computational precision and domain-specific functionality that external tools provide, highlighting the continued importance of tool-augmented frameworks in scientific problem-solving environments.

**Performance Varies Significantly Across Disciplines.** Models generally perform better on Physics and Chemistry than on Life Sciences and Materials Science. Life Sciences exhibits strong tool dependency: the average improvement is +2.5% for Physics, +7.0% for Chemistry, +3.7% for Materials Science, and +8.4% for Life Sciences, as summarized in Appendix A. Our analysis reveals that many Life Science tasks necessitate precise tool execution, such as database queries and specialized computations, which lie beyond the capabilities of pure parametric reasoning. An illustrative Life Sciences case study is provided in Appendix F.3.

### 6.3. RQ2: What Failure Patterns Emerge in Multi-step Scientific Tool-use?

We investigate failure mechanisms in multi-step scientific tool orchestration. Our analysis reveals weak models fall into invocation loops due to broken error-handling pipelines, resulting in fragile trajectory-level recovery dynamics.

**Excessive Tool Invocation Loop.** First, we observe a distinct efficiency gap. As shown in Figure 4 (left), there is a weak negative correlation ($r = -0.18$) between tool usage frequency and task success. For instance, Qwen3-VL-8B-Inst averages 16.55 tool calls yet achieves only 23.4% accuracy, whereas GPT-5 reaches 41.3% with merely 3.41 calls. Detailed analysis reveals that weaker models frequently enter repetitive tool-calling loops, re-invoking similar tools without reducing uncertainty (see Appendix F.4). In contrast, top-performing models exhibit high information yield per call, solving complex workflows with targeted, sparse invocations. Targeted scientific tool-use training mitigates this inefficiency. Our fine-tuned model reduces average tool calls while improving tool-augmented accuracy by about 7% , demonstrating that supervision on scientific trajectories enhances the capability to resolve problems via scientific tools while significantly curtailing redundant invocations.

**Breakdown in Process-Level Feedback.** To understand why models fall into these repetitive loops, we analyze 6,617 error instances to identify where the reasoning process breaks down. We measure recovery capability using four metrics: **Adaptation** (responsiveness to error), **Tuning** (parameter refinement), **Switching** (strategic pivoting), and **Loop Escape** ($1-$ rate of identical repetition). Detailed calculations are provided in Appendix C.3. As shown in Figure 4 (middle), we report the distribution of these four metrics across all models (where higher is better). Results reveal widespread failures across all stages: **Adaptation** shows a median response rate of only 32.9%, indicating that models ignore the majority of error signals; **Tuning** rates are even lower at 6.6%, suggesting models fail to diagnose and fix specific parameter errors; successful strategic **Switching**

*Table 4.* Training ablation results. We report tool-augmented (ReAct) Success Rate (SR, %) by subject. Arrows indicate change from Qwen3-VL-8B. **Key findings**: error trajectories are essential, generic tools cause negative transfer, and scientific skills transfer across domains.

| Model | Physics | Chemistry | Life Sci | Materials | Overall |
|---|---|---|---|---|---|
| Qwen3-VL-8B | 24.0 | 28.6 | 24.1 | 7.1 | 23.4 |
| *Ablation: Training Data Composition* | | | | | |
| Qwen3-VL-8B-OtherTools | 21.1 -2.9 | 21.0 -7.6 | 20.1 -4.0 | 3.7 -3.4 | 18.5 -4.9 |
| Qwen3-VL-8B-NoError | 30.5 +6.5 | 26.4 -2.2 | 26.7 +2.6 | 14.7 +7.6 | 26.6 +3.2 |
| *Ablation: Domain Transfer* | | | | | |
| Qwen3-VL-8B-Physics | 30.5 +6.5 | 31.5 +2.9 | 25.0 +0.9 | 17.1 +10.0 | 28.2 +4.8 |
| Qwen3-VL-8B-Chem | 24.8 +0.8 | **35.6** +7.0 | **31.2** +7.1 | **20.0** +12.9 | 28.2 +4.8 |
| Qwen3-VL-8B-Merged | **33.0** +9.0 | 35.2 +6.6 | 31.0 +6.9 | 9.1 +2.0 | **30.1** +6.7 |

occurs in only 15.3% of cases. Compounding these failures, the **Loop Escape** rate remains low at 35.7%. This chain of failures creates a critical bottleneck. Lacking the ability to interpret feedback or fix their actions, models inevitably revert to the repetitive tool usage described earlier.

**Trajectory-Level Recovery Dynamics.** Finally, we examine how error-handling capabilities change over long tasks. We define the Recovery Rate as the probability that a model successfully performs a correct action immediately after making a mistake. Measuring this across steps reveals two different patterns. As shown in Figure 4 (right), strong models display a *Rise-Fall-Rise* pattern: Claude-Sonnet-4's recovery rate increases from 40% (steps 2–3) to 57% (steps 6–7), drops to 9% (steps 8–9), then rebounds to 63% (steps 10–12). This pattern suggests that even top models encounter difficult phases mid-trajectory, but crucially, they can escape and restore effective error correction. Weaker models lack this resilience. Qwen3-VL-8B declines monotonically from 29% to 10% and remains low thereafter. Once these models enter error traps, they remain stuck. The critical differentiator is not whether models attempt incorrect tools, but whether they can break free from the resulting error loops. This recovery capability is fundamental for robust multi-step scientific tool-use.

### 6.4. RQ3: What Makes Scientific Tool-use Training Effective?

Building on evidence that fine-tuning significantly improves scientific tool-use capabilities, as shown in Figure 4(left), we now investigate *which training design choices* contribute to these gains through systematic ablation studies.

**Scientific tool-use is domain-specific yet transferable.** To test whether generic tool-use data can substitute for scientific tools, we fine-tune Qwen3-VL-8B on non-scientific tools (e.g., web search, file operations, calendar APIs). As shown in Table 4, this other-domain fine-tuning leads to a 4.9 point drop in overall score relative to the base model,

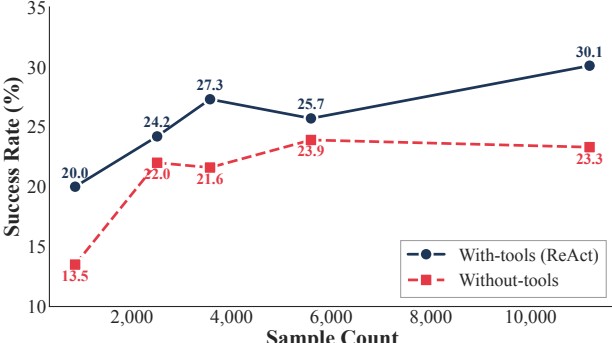

*Figure 5.* Scaling behavior: Tool-augmented (ReAct) performance improves with data size, while tool-free reasoning saturates early, highlighting the value of large-scale tool-use trajectories.

indicating negative transfer. In contrast, within the scientific tool space, we observe consistent cross-disciplinary transfer: models fine-tuned on a single discipline demonstrate improved performance in other scientific domains. Our detailed analysis in Appendix F.2 reveals that unlike generic baselines, scientific tool tuning instills rigorous paradigms such as constraint checking and numerical precision. This confirms that scientific data is indispensable for cultivating both domain-grounded behaviors and transferable meta-skills that generalize across disciplines.

**Error recovery trajectories are essential.** We compare Qwen3-VL-8B-Merged trained on full trajectories with error recovery to Qwen3-VL-8B-NoError trained only on clean trajectories. Table 4 shows that Qwen3-VL-8B-NoError underperforms across most subjects. This performance gap indicates that exposure to failure-correction sequences is important for robust tool-use.

**Tool-use capabilities scale more readily than static SFT knowledge.** Finally, we examine how performance scales with training data under tool-augmented versus tool-free evaluation. Figure 5 tracks success rates across training checkpoints. Tool-augmented Success Rate improves with data scale, despite non-monotonic fluctuations, while tool-free performance saturates early and plateaus despite continued training. This contrast suggests that tool-use behaviors are more data-scalable: additional trajectories teach models better interaction patterns, verification habits, and error handling. In contrast, the static knowledge and reasoning acquired through SFT alone appears harder to scale with more data. These findings support the value of large-scale tool-augmented training data for building capable scientific agents, as tool integration enables continued performance gains that are beyond the reach of internal textual reasoning.

# 7. Conclusion

In this work, we introduce **SciAgentGym** and **SciAgent-Bench**, shifting scientific evaluation from static knowledge to dynamic, tool-augmented reasoning. Analysis reveals that while tools are essential, current models struggle with long-horizon workflows, often failing to recover from errors. Thus, we propose **SciForge**, an execution-grounded synthesis method generating logic-aware data from real interactions. Training on these verified trajectories enables our 8B model to outperform 200B+ baselines. By providing reproducible infrastructure and scalable synthesis, we hope this work can lay the foundation for future scientific agents.

# Acknowledgements

The authors wish to thank the anonymous reviewers for their helpful comments. This work was partially funded by the Shanghai Municipal Special Program for Basic Research on General AI Foundation Models (Grant No. 2025SHZDZX025G07), in collaboration with Shanghai Artificial Intelligence Laboratory, and National Natural Science Foundation of China (No.62521004, 62476061, 62576106, 62376061).

# Impact Statement

This paper presents work whose goal is to advance the field of Machine Learning. There are many potential societal consequences of our work, none of which we feel must be specifically highlighted here.

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

## A. Per-Discipline Score Breakdown

Table 5 reports a complete per-discipline breakdown of success rates for both **with tools (ReAct)** and **without tools** settings. Due to a small number of unresolved non-response cases after repeated reruns, success rates are computed over the effective evaluated set for each model and subset.

## B. SciAgentGym Environment Details

### B.1. Tool-Type Distribution by Discipline

Figure 6 summarizes how the four tool categories (numerical computation, data processing, visualization, database queries) are allocated across the six scientific disciplines in SciAgentGym, complementing the main-text discussion of tool construction.

### B.2. Source Benchmarks

SciAgentBench aggregates tasks from multiple established scientific benchmarks; Table 6 summarizes the included sources and their licenses. We ensure that SciAgentBench was constructed in compliance with the licensing agreements of all aggregated scientific benchmarks and with full respect for the intellectual property rights of the original contributors. We believe our aggregated framework does not cause any potential risks."

### B.3. Difficulty Filtering Criteria

We apply a filtering pipeline that (i) evaluates each candidate task under zero-shot prompting with four frontier LLMs (Claude-Sonnet-4.5(Anthropic, 2025), GPT-5(Singh et al., 2025), DeepSeek-R1(Guo et al., 2025a), and Qwen-235B(Yang et al., 2025), (ii) computes ensemble accuracy and retains only tasks whose mean accuracy is below 50%, (iii) performs stratified sampling to preserve the original domain distribution, and (iv) verifies that the remaining tasks are executable within the SciAgentGym environment.

### B.4. Tool Signature and Serialization Specifications

As defined in Eq. (1), each tool $v \in V_d$ specifies a mapping between scientific types. To make this abstract signature executable under LLM function-calling interfaces, we enforce a serialization protocol that constrains both inputs and out-

*Table 5.* Per-discipline success rates (SR, %) for **with tools (ReAct)** and **without tools**.

| Model | w/ Tools (ReAct) | | | | | w/o Tools | | | | |
|---|---|---|---|---|---|---|---|---|---|---|
| | **Phys.** | **Chem.** | **Mat.** | **Life** | **Overall** | **Phys.** | **Chem.** | **Mat.** | **Life** | **Overall** |
| *Closed-Source Models* | | | | | | | | | | |
| GPT-5 | 46.3 | 43.8 | 28.6 | 32.3 | 41.3 | 37.6 | 35.0 | 25.0 | 15.6 | 32.3 |
| Grok-4-1 | 47.2 | 38.2 | 32.4 | 30.0 | 40.3 | 36.8 | 30.0 | 31.0 | 9.4 | 30.4 |
| Claude-Sonnet-4 | 39.4 | 39.5 | 27.0 | 25.0 | 35.9 | 25.7 | 22.2 | 13.5 | 21.9 | 22.4 |
| Gemini-2.5-Flash | 38.3 | 32.4 | 28.6 | 17.2 | 32.7 | 36.4 | 29.1 | 19.4 | 9.7 | 28.5 |
| Gemini-2.5-Pro | 37.3 | 35.1 | 26.5 | 18.8 | 32.6 | 28.7 | 24.7 | 24.3 | 12.5 | 24.8 |
| Gemini-2.5-Pro-Think | 33.3 | 28.9 | 21.2 | 21.9 | 28.8 | 38.3 | 26.2 | 24.3 | 9.4 | 28.9 |
| O3 | 35.5 | 37.3 | 32.4 | 6.5 | 32.0 | 33.0 | 27.2 | 18.9 | 12.5 | 26.6 |
| O4-mini | 31.2 | 35.5 | 30.6 | 20.0 | 31.1 | 33.9 | 28.4 | 21.6 | 12.5 | 27.8 |
| GPT-4o | 21.3 | 20.5 | 8.6 | 16.0 | 18.7 | 25.9 | 14.8 | 5.4 | 6.2 | 17.1 |
| *Open-Source Large Models (>30B)* | | | | | | | | | | |
| GLM-4.6V | 30.9 | 37.5 | 22.2 | 18.8 | 30.9 | 34.3 | 26.6 | 14.3 | 9.4 | 26.0 |
| Qwen3-VL-235B-Think | 30.6 | 29.5 | 22.9 | 22.6 | 28.0 | 30.0 | 27.9 | 10.3 | 13.3 | 24.4 |
| Qwen3-VL-235B-Inst | 28.1 | 26.5 | 5.0 | 17.2 | 23.9 | 31.5 | 20.0 | 18.9 | 6.2 | 23.0 |
| Qwen3-VL-32B-Think | 33.0 | 31.2 | 8.8 | 22.6 | 27.9 | 32.0 | 25.6 | 12.1 | 9.4 | 24.4 |
| Qwen3-VL-32B-Inst | 31.8 | 29.3 | 20.0 | 16.1 | 27.4 | 29.4 | 19.8 | 13.5 | 18.8 | 22.8 |
| *Open-Source Small & Medium Models (≤30B)* | | | | | | | | | | |
| Qwen3-VL-8B-Inst | 24.0 | 28.6 | 7.1 | 24.1 | 23.4 | 27.1 | 13.8 | 5.6 | 15.6 | 18.4 |
| SciAgent-8B | 33.0 | 35.2 | 9.1 | 31.0 | 30.1 | 25.7 | 30.4 | 8.1 | 15.6 | 23.3 |
| Qwen3-VL-4B-Inst | 23.8 | 20.6 | 10.3 | 13.3 | 19.7 | 21.5 | 17.9 | 5.6 | 12.5 | 17.0 |
| SciAgent-4B | 28.4 | 28.4 | 14.7 | 19.4 | 25.2 | 15.6 | 24.7 | 10.8 | 12.5 | 17.4 |
| Pixtral-12B | 7.5 | 6.3 | 5.9 | 10.0 | 7.2 | 11.0 | 6.2 | 8.6 | 0.0 | 7.8 |
| *Average* | 31.6 | 30.8 | 19.0 | 20.1 | 28.3 | 29.2 | 23.7 | 15.3 | 11.7 | 23.3 |

| No. | Source | License |
|---|---|---|
| 1 | **SciInstruct** (Zhang et al., 2024) | CC BY 4.0 |
| 2 | **GPQA** (Rein et al., 2023) | CC BY 4.0 |
| 3 | **BMMR** (Xi et al., 2025c) | Apache-2.0 |
| 4 | **SFE** (Zhou et al., 2025) | MIT License |
| 5 | **RBench-V** (Guo et al., 2026) | Apache-2.0 |

*Table 6.* Source benchmarks used to construct SciAgentBench.

puts to JSON-compatible representations. Complex scientific objects are reconstructed *within* tools from serializable identifiers (e.g., SMILES, POSCAR, file paths, database IDs), while outputs follow a unified dictionary schema augmented with explicit scientific metadata (e.g., units, status). Table 7 summarizes the hard requirements used in our implementation (distilled from the system/construction prompts), covering validation, large-data handling, and traceability.

### B.5. Details of Taxonomy Organization

To improve tool discoverability and composability in multi-step tool-chains, we organize tools along two orthogonal axes: *function* and *granularity*. The function axis groups tools by their primary role in a scientific workflow (query, computation, analysis, visualization), while the granularity axis distinguishes atomic primitives from composite operations. This taxonomy is consistent with our layered architecture (Atomic → Composite → Visualization). Table 8 summarizes definitions for each tool type and provides a representative example function.

## C. Benchmark Construction Details

### C.1. Benchmark Construction

We reuse the same source-benchmark pool and curation pipeline as SciAgentGym for raw task collection . All components of the pipeline remain identical to Appendix B, except for the difficulty filtering stage, where we replace the original filtering model with GPT-5(Singh et al.,

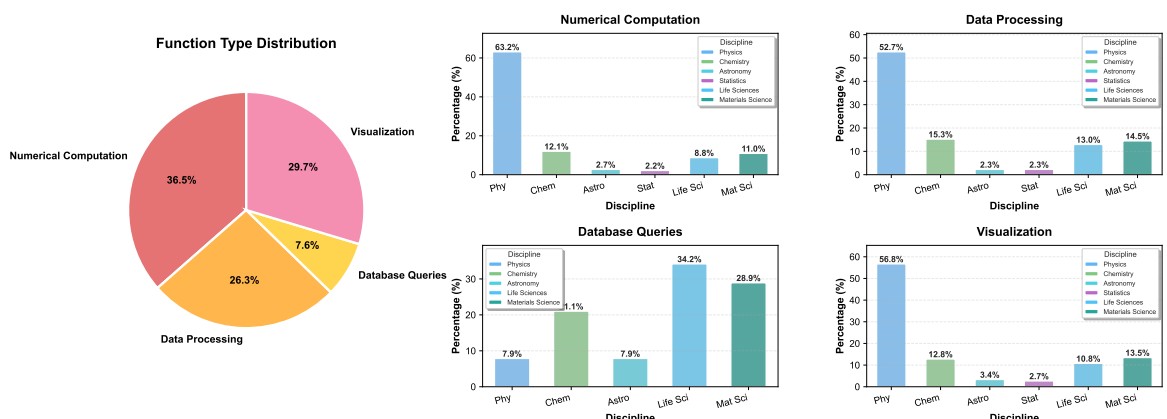

*Figure 6.* Proportional distribution of four tool categories (numerical computation, data processing, visualization, database queries) across Physics, Chemistry, Astronomy, Statistics, Life Sciences, and Materials Science. The Physics, Astronomy, and Statistics disciplines constitute the Physics domain mentioned in the main text.

---

*Implementation Specifications for Tool Signatures and Serialization*

**1. Input Serialization ($\alpha_i^v$ Handling).** All inputs must be JSON-serializable primitives (`str`, `int`, `float`, `bool`) or standard collections (`List`, `Dict`).
- **Internal Construction:** Complex objects (e.g., `rdkit.Chem.Mol`, `pymatgen.Structure`, `scipy.sparse.csr_matrix`) must be reconstructed *inside the tool* from serializable identifiers such as SMILES strings, POSCAR text, file paths, or database IDs.
- **Boundary Checks:** Each tool must validate types, ranges, and special cases (e.g., zero/extreme values) before executing scientific logic.

**2. Output Encapsulation ($\beta_j^v$ Handling).** All results must be returned in a unified dictionary schema to preserve scientific context.
- **Standard Return:** `{'result': main_value, 'metadata': {...}}` (e.g., units, status flags, data sources).
- **Large/Non-serializable Data:** High-dimensional or non-serializable outputs (e.g., sparse matrices) must be persisted under `./mid_result/{subject}/` (e.g., physics/chemistry/materials). The returned summary must include `filepath`, `shape`, `nnz`, and `sparsity`.

**3. Quality and Traceability Requirements.**
- **Type Hints:** All tools must provide complete Python type hints for parameters and return values.
- **Scientific Metadata:** Metadata should include units and relevant diagnostic information (e.g., convergence status, databases used) to support reproducible tool-chains.

*Table 7.* Executable interface specifications for SciAgentGym tools: input serialization requirements, output encapsulation schema, and traceability constraints for compatibility with LLM function-calling interfaces.

---

2025), Claude-Sonnet-4.5(Anthropic, 2025), Gemini-2.5-Pro(Team et al., 2023), and DeepSeek-R1(Guo et al., 2025a), as detailed in Appendix B.3

We construct our benchmark with two subsets: a unimodal dataset and a multimodal dataset, where each question may be paired with one or more images. For every instance, we provide both a reference solution and a structured intermediate decomposition, enabling the evaluation of multi-step reasoning beyond final-answer correctness. Crucially, both subsets are annotated with expert tool-use trajectories that define the intended sequence of tool invocations required to solve each problem. When applicable, we further include canonical tool inputs and their corresponding outputs, allowing for reproducible tool execution and fine-grained supervision of tool-augmented reasoning. This trajectory-

centric design supports not only success-based evaluation but also efficiency-aware metrics, such as SPL (Success weighted by Path Length), which weights successful solutions by their tool-use length relative to the expert reference. Figure 8 illustrates a representative expert trajectory.

### C.2. Interpretation Note.

We acknowledge that scientific problem-solving often admits multiple valid solution paths. Our golden traces represent *one* expert-verified execution strategy, not necessarily the unique optimal solution. Consequently, SPL measures **alignment with the reference strategy** rather than absolute planning optimality. An agent achieving lower SPL may have taken an alternative valid approach rather than an inefficient one. We retain SPL as it provides a consistent,

| *Tool Taxonomy: Organization by Function and Granularity* |
|---|

**1. Function Axis (Workflow Role).** Tools are organized by their primary role in the scientific workflow: *query*, *computation*, *analysis*, and *visualization*.
- **Query:** Retrieve hierarchical facts/records from external resources or local indices and return normalized fields for downstream steps.
  - **Example:** `fetch_property_from_database(identifier, property_name)`
- **Computation:** Execute core scientific calculations or model components under strict boundary checks, producing unit-aware results and/or persisted artifacts when needed.
  - **Example:** `construct_hamiltonian(params, ...)`
- **Analysis:** Perform post-processing and higher-level interpretation over computed/retrieved artifacts (e.g., diagnostics, aggregation, comparisons, error/uncertainty handling).
  - **Example:** `analyze_commutator(matrix_a, matrix_b)`
- **Visualization:** Generate domain-specific scientific figures; the tool must persist figures and return references to generated artifacts for downstream use.
  - **Example:** `visualize_domain_specific(data, vis_type, ...)`

**2. Granularity Axis (Atomic to Composite).** Tools are further organized by operational granularity, from *atomic primitives* to *composite operations*.
- **Atomic primitives:** Single-responsibility functions with minimal side effects and stable interfaces; intended as reusable building blocks.
- **Composite operations:** Higher-level procedures that orchestrate multiple atomic primitives into a complete workflow (not simple concatenation), while preserving traceable intermediate states.

*Table 8.* Tool taxonomy used in SciAgentGym, organizing tools along two axes: workflow function (query, computation, analysis, visualization) and operational granularity (atomic primitives vs. composite operations), with representative function examples.

reproducible baseline for cross-model comparison, while recognizing this limitation in interpretation.

## C.3. Feedback Utilization Failure Taxonomy

**Adaptation.** Whether the model changes its action after receiving an error signal.

$$\text{Adaptation} = P(a_{t+1} \neq a_t \mid y_t = \text{fail}) \qquad (8)$$

**Tuning.** Whether the model successfully fixes the error by retrying the same tool with corrected inputs.

$$\text{Tuning} = P(y_{t+1} = \text{success} \mid \mathcal{C}_{\text{tune}}) \qquad (9)$$

where $\mathcal{C}_{\text{tune}} = \{y_t = \text{fail}, a_{t+1} \neq a_t, tool_{t+1} = tool_t\}$.

**Switching.** Whether the model successfully resolves the error by switching to a different tool or reasoning strategy.

$$\text{Switching} = P(y_{t+1} = \text{success} \mid \mathcal{C}_{\text{switch}}) \qquad (10)$$

where $\mathcal{C}_{\text{switch}} = \{y_t = \text{fail}, a_{t+1} \neq a_t, tool_{t+1} \neq tool_t\}$.

**Loop Escape.** Whether the model avoids repeating the exact same failed action.

$$\text{Loop Escape} = 1 - P(a_{t+1} = a_t, y_{t+1} = \text{fail} \mid y_t = \text{fail}) \qquad (11)$$

**Interpretation.** These four metrics form a diagnostic pipeline for feedback utilization:

- **Adaptation** tests whether the model *notices* the error and attempts any change.

- **Tuning** tests whether the model can *fix* the error using the same tool.

- **Switching** tests whether the model can *circumvent* the error via alternative strategies.

- **Loop Escape** tests whether the model *avoids* falling into repetitive failure patterns.

## C.4. Data Annotation and Quality Review Process

To ensure reliability and evaluability during benchmark construction, we conduct human quality review on all samples , verifying clarity and unambiguity, the logical correctness of step-by-step reasoning and tool-call ordering, the consistency of tool outputs via recomputation, the correctness of final answers, and compliance with the benchmark template. Each item receives at least one full end-to-end check, and complex cases are double-reviewed; workloads are distributed over a 30–60 day period to reduce fatigue-induced errors.

## D. Execution-Grounded Synthesis

### D.1. Algorithm Details

A complete pseudocode description of the backward program construction process is presented in Algorithm 1.

**Algorithm 1** Backward Program Construction

**Require:** Target tool $v_{\text{goal}}$, graph $\mathcal{G}_d = (\mathcal{V}_d, \mathcal{E}_d)$, max depth $D_{\text{max}}$, exploration rate $\epsilon$
**Ensure:** Executable program graph $\mathcal{P} = (\mathcal{V}_\mathcal{P}, \mathcal{B})$
1: $\mathcal{V}_\mathcal{P} \leftarrow \{v_{\text{goal}}\}, \mathcal{B} \leftarrow \emptyset$
2: $\text{anc}[v_{\text{goal}}] \leftarrow \emptyset, \text{depth}[v_{\text{goal}}] \leftarrow 0$
3: $\text{queue} \leftarrow [(v_{\text{goal}}, 0)]$
4: **while** $\text{queue} \neq \emptyset$ **do**
5:    $(v, d) \leftarrow \text{queue.popfront()}$
6:    **for** each input slot $j \in [k_v]$ **do**
7:       $\mathcal{A} \leftarrow \text{anc}[v] \cup \{v\}$ {Path ancestors}
8:       $\mathcal{C} \leftarrow \{(u, i) \in \mathcal{O}(v, j) : u \notin \mathcal{A}, d < D_{\text{max}}\}$
9:       **if** $\mathcal{C} \neq \emptyset$ **then**
10:          $\mathcal{C}^* \leftarrow \{(u, i) \in \mathcal{C} : \text{stage}(u) \leq \text{stage}(v)\}$ {Stage-compliant candidates}
11:          $u^* \leftarrow \text{UniformSample}(\mathcal{C}^*)$ if $\mathcal{C}^* \neq \emptyset$ else $\text{UniformSample}(\mathcal{C})$
12:          $(u, i) \leftarrow \text{EpsilonGreedy}(\mathcal{C}, u^*, \epsilon)$ {$\epsilon$-greedy sampling}
13:          $\mathcal{B} \leftarrow \mathcal{B} \cup \{(u, i) \rightarrow (v, j)\}$
14:          **if** $u \notin \mathcal{V}_\mathcal{P}$ **then**
15:             $\mathcal{V}_\mathcal{P} \leftarrow \mathcal{V}_\mathcal{P} \cup \{u\}, \text{depth}[u] \leftarrow d + 1$
16:             $\text{anc}[u] \leftarrow \mathcal{A}, \text{queue.append}((u, d+1))$
17:          **end if**
18:       **else**
19:          $\mathcal{B} \leftarrow \mathcal{B} \cup \{\mathcal{R}_{\alpha_j^v} \rightarrow (v, j)\}$ {Root initializer}
20:       **end if**
21:    **end for**
22: **end while**
23: **return** $\mathcal{P} = (\mathcal{V}_\mathcal{P}, \mathcal{B})$

# E. Experimental Settings

## E.1. Evaluation Details

We evaluate OpenAI-compatible chat-completion models under a ReAct-style protocol, comparing **with tools** and **without tools** settings. In the **with tools** setting, only task-relevant tools are exposed, each defined via the OpenAI tool schema (name, description, JSON-schema parameters). The model follows a system prompt enforcing a two-stage *Planning → Execution* procedure with explicit *Thought–Action–Observation* loops (Table 9); for the reverse setting, a convergence prompt enforces strict JSON outputs (Table 10). In the **without tools** setting, tools are disabled and the model outputs a reasoning trace followed by a boxed final answer (Table 12).

**Inference and Decoding Configuration.** Unless otherwise stated, all models are evaluated using a temperature of 0.7, following standard deployment settings. For tool-enabled runs, we set `tool_choice = "auto"` and `parallel_tool_calls = false`, cap tool interac-

tions at 50 rounds per attempt, and enforce a per-request timeout of 300 seconds. To ensure robust evaluation against strict formats, we employ a final answer normalization step. As detailed in Table 11, this prompts the model to convert its final response into a strict, task-specific JSON template, ensuring that numeric literals and boolean values adhere to the required schema for automated scoring.

**Metrics and Verification.** We evaluate outputs using *strict hierarchical accuracy*, recursively matching the predicted JSON against ground truth with a numeric tolerance of 0.05. To mitigate brittleness in long-form textual fields, we employ an LLM-based semantic verification step using `gpt-4.1` whenever strict matching fails solely due to textual discrepancies. The verifier operates under a system prompt that enforces binary scoring through a strict rubric-based evaluation framework.Additionally, specific judge prompts for semantic equivalence and binary correctness checks are detailed in Table 13.

## E.2. Training details

This appendix provides details on our fine-tuning setup, including trace generation, hyperparameters, and runtime safeguards, using the same prompt structures described in the evaluation settings.

**Training Trace Generation.** For supervised fine-tuning, we generate refined tool-use traces using a function-calling agent empowered to autonomously manage tool invocations (`tool_choice="auto"`). To ensure robust reasoning, generation is performed with a temperature of 0.3, capping interactions at 50 rounds per attempt. This is immediately followed by the normalization stage (temperature 0.1) detailed in Table 11, which coerces the final output into a strict, task-specific JSON format. The resulting trajectories—comprising serialized `tool_call` and `tool_response` messages along with the normalized answer—serve as the training targets.

**Supervised Fine-tuning (SFT).** We fine-tune the `Qwen3-VL-8B-Instruct` backbone using full-parameter SFT for 3 epochs. The training infrastructure leverages DeepSpeed ZeRO-3, gradient checkpointing, and FlashAttention to maximize efficiency. To maintain the pre-trained multimodal alignment stability, we explicitly freeze the vision backbone and the projector modules (`freeze_vit=true`, `freeze_aligner=true`), updating only the language model parameters. The SFT setup uses a learning rate of $1 \times 10^{-6}$ in `bfloat16` precision, with a maximum sequence length of 16384, batch size 2 per device, and gradient accumulation of 4 on 8 GPUs.

---

*Forward **with tools** Prompt Composition*

---

**System Prompt (ReAct).** You are a top-tier AI scientific assistant. Your task is to solve complex, multimodal, multi-step problems from domains such as physics, chemistry, and biology.

**Task Instructions: Two-Stage Method.** You must strictly follow the two stages below to answer the question:
- **Stage 1: Planning.** Before executing any actions, first generate a high-level, step-by-step plan outlining how you will break down the problem, which tools you will use, and the dependencies among tool calls.
- **Stage 2: Execution.** After providing the plan, execute it step by step. Strictly follow the ReAct format (**Thought**, **Action**, **Observation**) until you reach the final answer.

**Output Format Specification.**
[**Stage 1: Planning**]
**Plan:**
1. *[Your plan step 1, e.g., analyze the molecular structure in the input image]*
2. *[Your plan step 2, e.g., use* `analyze_molecule` *to obtain SMILES]*
3. *[Your plan step 3, e.g., pass SMILES into* `calculate_properties`*]*
4. *[. . . ]*
5. *[Your plan step N, e.g., summarize all properties and answer the question]*
[**Stage 2: Execution**]
**Thought:** Describe your current thought, which plan step you are on, and why you need to call this specific tool.
**Action:** Please issue a tool call in this step.
*(After you submit Action, you will receive an Observation)*
**Observation:** The evaluation system will insert the tool output here.
*(Repeat the loop as needed.)*
**Thought:** I have collected all necessary information and can now generate the final answer.
**Final Answer:** Here is the final, complete answer to the original question.

**User Message (Original Question + Appended Answer-Only Constraint).**
**Original question:** {`original_question_text`}

You should strictly respond in this exact format and answer the question in its original language:
`###Answer###`
The final answer wrapped in LaTeX boxed format: | final answer |

---

*Table 9.* In forward tool mode (testallforfailed), the ReAct instruction is provided as a system prompt, while the user message is constructed by concatenating the original question with an appended answer-only format constraint.

**Runtime Safeguards and Prompt Transparency.** To prevent resource exhaustion during both training data generation and evaluation, we enforce a strict timeout hierarchy: batch jobs are limited to 1200 seconds per task with up to 3 retries and a 5-second backoff, while individual shell commands and tool executions are hard-capped at 60 and 30 seconds, respectively. Regarding reproducibility, all experiments utilize the exact system instructions and judge prompts presented in the Evaluation Details section, with sensitive credentials redacted from public artifacts.

## F. Case study

### F.1. Thin-Film Interference: Full Trace vs. Expert Reference

For thin-film interference, we juxtapose an end-to-end interaction trace that contains partial tool-call failures (Figure 7) with the corresponding expert reference workflow (Figure 8), which integrates analytic derivation and numerical/visual verification.

### F.2. Controlled Model Comparison on the Same Problem Instance

Figure 9 contrasts two Qwen3-VL-8B variants on the same truss permissible-load task. The tool-tuned variant satisfies key *task constraints* by explicitly evaluating both force directions and returning a complete two-part answer, while reporting numerically stable values consistent with the tool outputs. By comparison, the generic baseline expends additional calls on auxiliary steps yet violates the output constraint by omitting required parts, despite having computed relevant intermediate quantities.

### F.3. Life-Science Case Study

Figure 10 provides a representative Life Sciences trace in which solving the plasmid-replacement task requires *precise tool execution*—notably database-backed queries for plasmid properties (e.g., copy number, origin, resistance) and a downstream difficulty/strategy computation—before a valid protocol-level plan can be composed.

---

*Reverse **with tools** Prompt Composition*

---

**System Prompt (ReAct).** You are a top-tier AI scientific assistant. Your task is to solve complex, multimodal, multi-step problems from domains such as physics, chemistry, and biology.

**Task Instructions: Two-Stage Method.** You must strictly follow the two stages below to answer the question:
- **Stage 1: Planning.** Before executing any actions, first generate a high-level, step-by-step plan outlining how you will break down the problem, which tools you will use, and the dependencies among tool calls.
- **Stage 2: Execution.** After providing the plan, execute it step by step. Strictly follow the ReAct format (**Thought**, **Action**, **Observation**) until you reach the final answer.

**Output Format Specification.**
[Stage 1: Planning]
**Plan:**
1. *[Your plan step 1, e.g., analyze the molecular structure in the input image]*
2. *[Your plan step 2, e.g., use* `analyze_molecule` *to obtain SMILES]*
3. *[Your plan step 3, e.g., pass SMILES into* `calculate_properties`*]*
4. *[...]*
5. *[Your plan step N, e.g., summarize all properties and answer the question]*
[Stage 2: Execution]
**Thought:** Describe your current thought, which plan step you are on, and why you need to call this specific tool.
**Action:** Please issue a tool call in this step.
*(After you submit Action, you will receive an Observation)*
**Observation:** The evaluation system will insert the tool output here.
*(Repeat the loop as needed.)*
**Thought:** I have collected all necessary information and can now generate the final answer.
**Final Answer:** Here is the final, complete answer to the original question.

**Final User Message (Post-Interaction JSON Convergence Prompt + Answer Template).**
Now output **only** strict JSON. It must exactly match the following structure (key names, hierarchy, and all fields must be present). All numeric values must be numbers (keep 2–3 decimals), booleans must be `true`/`false`, and strings must contain explanatory text. Do not output any explanatory text or any prefix/suffix. Do not include any extra fields. If you cannot compute a value, provide a parseable approximate value.

**Answer template:** {`answer_template_json_here`}

---

*Table 10.* In the reverse setting (testall), the model first runs under the ReAct system prompt during the tool interaction. After the interaction ends, a final user prompt is appended to force template-compliant JSON output for field-level matching/scoring.

---

*User Prompt: Normalization / Finalization*

---

**Instruction.** Output **only** a strict JSON object that exactly matches the provided template (keys, nesting, and required fields must be identical). Use numeric literals for numeric values (keep 2–3 decimals), `true`/`false` for booleans, and strings for textual fields. Do not output any additional text, explanations, or extra keys. If a value cannot be computed precisely, provide a reasonable approximation while maintaining valid JSON syntax.

**Template.** `<answer_template JSON>`

---

*Table 11.* Normalization prompt used to convert the final response into a strict task-specific JSON template.

---

*Without Tools Prompt Composition (No Tools)*

---

**User Message (Original Question + Appended Format Constraint).**
**Original question:** {`original_question_text`}

You should strictly respond in this exact format and answer the question in its original language:

`###Reasoning Process###`
*[Your step-by-step reasoning process here]*
`###Answer###`
The final answer wrapped in LaTeX boxed format:
$\boxed{\text{final answer}}$

---

*Table 12.* In **without tools**, no ReAct system prompt is used; the user message is constructed by concatenating the original question with the required output format (reasoning + boxed answer).

## F.4. Failure Modes

We observe two recurring failure patterns in tool-augmented scientific reasoning. One is *degenerate tool-use loops*: in Case 67, the model repeatedly re-invokes the same shear-stress subroutine without updating the governing torque or applying the allowable-stress constraints, eventually exhausting the round budget (Figure 11). A closely related degeneration appears in the mass-spectrometry setting:

Case 53 shows that weaker models may either fail to follow the required tool-call format or collapse into repetitive peak-extraction attempts with identical outputs, and later even degrade to invalid empty arguments that trigger validation errors (Figure 12).

| *Judge Prompt: LLM-Based Verification* |
|---|

**Semantic equivalence check.** Determine whether two answers are semantically equivalent. Even if the wording differs, mark them as a match if the core meaning, numbers, and logical relationships are consistent. Output **only** MATCH or NO_MATCH.
**Expected:** `<expected>`
**Actual:** `<actual>`

**Binary correctness judge.** Given a question, a reference answer, and a model answer, determine whether the model answer is correct. Judge strictly against the reference; paraphrases are allowed if the reasoning and results are consistent. Output **only** CORRECT or INCORRECT.
**Question:** {`question_text`}
**Reference:** {`standard_answer`}
**Model answer:** {`model_answer`}

*Table 13.* Judge prompt templates used for secondary verification.

The second pattern is *formula-level errors* that survive despite otherwise plausible reasoning. In Case 68, a representative failure produces a structurally incorrect closed form (e.g., an unnecessary functional transformation) while the correct expression is comparatively simple, illustrating that "near-miss" analytical mistakes remain a major source of incorrect answers (Figure 13).

**Case 05 – claude-sonnet-4-20250514**

**Problem:** White light is incident normally on a thin film with refractive index $n_2 = 1.4$. The film thickness is $d = 350$ nm. Given that $n_1 > n_2$ and $n_2 < n_3$, determine which visible wavelengths are enhanced in the reflected light.

**Input image**:

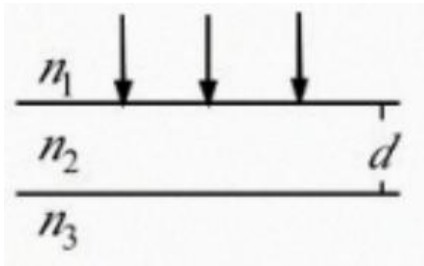

**Full interaction trace**

**R1 Act:** `calculate_thin_film_interference(n1=1.5, n2=1.4, n3=1.6, d=350)`
**Obs 1:** ERROR: Object of type function is not JSON serializable

**R2 Act:** `find_extrema_wavelengths(n1=1.5, n2=1.4, n3=1.6, d=350)`
**Obs 2:** enhanced_wavelengths = [391.85, 653.33] nm; weakened_wavelengths = [490.0] nm

**R3 Act:** `optical_path_difference(n2=1.4, d=350, wavelength=653.33)`
**Obs 3:** OPD = 980.0 nm; phase_diff = 9.42; interference = `partial`

**R4 Act:** `optical_path_difference(n2=1.4, d=350, wavelength=391.85)`
**Obs 4:** OPD = 980.0 nm; phase_diff = 15.71; interference = `partial`

**R5 Act:** `reflectance_spectrum(n1=1.5, n2=1.4, n3=1.6, d=350, wavelength_range=[\"380\",\"750\"])`
**Obs 5:** ERROR: DTypePromotionError (wavelength_range parameter type error)

**R6 Act:** `reflectance_spectrum(n1=1.5, n2=1.4, n3=1.6, d=350, wavelength_range=[\"380nm\",\"750nm\"])`
**Obs 6:** ERROR: DTypePromotionError (wavelength_range parameter type error)

**Final answer + evaluation:**

Model final answer: $\boxed{392 \text{ nm, } 653.33 \text{ nm}}$

Judge: is_correct = True (standard = 653.33 nm and 392 nm)

*Figure 7.* Case 05: Full tool-augmented reasoning trace for thin-film interference with partial tool-call failures.

---

**Expert Trace**

**Problem:** White light is incident normally on a thin film with refractive index $n_2 = 1.4$. The film thickness is $d = 350$ nm. Given that $n_1 > n_2$ and $n_2 < n_3$, determine which visible wavelengths are enhanced in the reflected light.

**Input image:**

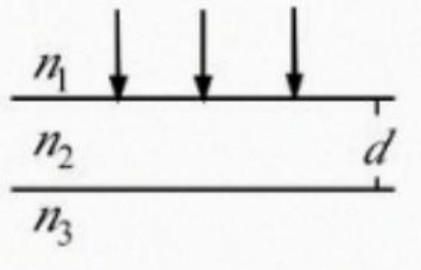

---

**Expert multi-round dialogue**

**R1 Expert note:** Determine the reflection phase inversions.

**Obs 1:** Exactly one phase inversion occurs (one half-wave loss), so constructive reflection satisfies

$$2n_2 d = \left(m + \frac{1}{2}\right)\lambda, \qquad \lambda = \frac{2n_2 d}{m + \frac{1}{2}}.$$

Here, $2n_2 d = 2 \times 1.4 \times 350 = 980$ nm.

---

**R2 Act:** `calculate_thin_film_interference(n1=1.5, n2=1.4, n3=1.8, d=350, wavelength_range=[380,750], incidence_angle=0)`

**Obs 2:** enhanced_wavelengths = [392.0, 653.33] nm; weakened_wavelengths = [490.0] nm; condition: $2n_2 d = \left(m + \frac{1}{2}\right)\lambda$

---

**R3 Act:** `find_extrema_wavelengths(n1=1.5, n2=1.4, n3=1.8, d=350, wavelength_range=[380,750], num_points=3000)`

**Obs 3:** numeric peaks at [391.97, 653.27] nm; numeric minimum at [490.05] nm

---

**R4 Act:** `visualize_reflection_spectrum(n1=1.5, n2=1.4, n3=1.8, d=350, wavelength_range=[380,750], num_points=1000)`

**Obs 4:** output image path = `images/reflection_spectrum_xxx.png`; peaks marked near [391.9, 653.3] nm

**Output Image:**

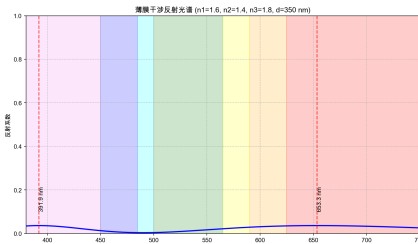

---

**Final answer:**

Enhanced visible wavelengths in reflection are approximately 392 nm (violet) and 653 nm (red).

**Final Answer:** $\boxed{392\,\text{nm}, \ 653\,\text{nm}}$

---

*Figure 8.* Case 05: Expert reference trace for thin-film interference, integrating analytical derivation, numerical verification, and visualization.

**Model Comparison (English) – Case 66 (Truss Permissible Load)**

**Problem:** The square truss structure shown in the figure consists of five steel rods with circular cross-sections, and all connections are hinges. Each rod has a diameter $d = 40\,\mathrm{mm}$, $a = 1\,\mathrm{m}$ and material of Q235 steel. $E = 200\,\mathrm{GPa}$, $[n]_{\mathrm{st}} = 1.8$. Solve: (1) Determine the permissible load of the structure; (2) If the direction of force $F_P$ is opposite to that in (1), will the permissible load change? If there is a change, what should it be?

**Reference answer:** Part 1: $189.6\,\mathrm{kN}$. Part 2: Yes; $68.9\,\mathrm{kN}$.

**Input image**

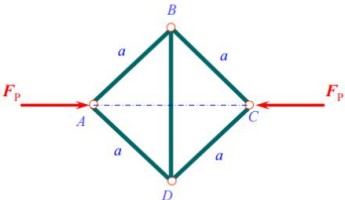

---

**Phase 1: Environment setup (shared)**

Registered tools: `determine_truss_allowable_load`, `calculate_cross_section_area`, `calculate_moment_of_inertia`, `visualize_truss_structure`

---

**Phase 2: Per-model interaction trace + evaluation**

**Model A (correct):** `Qwen3-VL-8B-Merged`

(Rounds=3; tool calls=2; both directions evaluated.)

**R1 Act:** `determine_truss_allowable_load(force_direction=inward)`
**Obs 1:** allowable external load $\approx 68.90\,\mathrm{kN}$; critical member `BD` (compression)

---

**R2 Act:** `determine_truss_allowable_load(force_direction=outward)`
**Obs 2:** allowable external load $\approx 194.89\,\mathrm{kN}$; critical member `AB` (compression)

---

**Final answer (translated from the model output):**
Permissible load (direction as in Part 1): $68.90\,\mathrm{kN}$. Reversing $F_P$ changes the permissible load to $194.89\,\mathrm{kN}$.
**Judge:** `is_correct=True`

---

**Model B (wrong):** `Qwen3-VL-8B-OtherTools`

(Rounds=8; tool calls=7; computes both directions but collapses to a partial final answer.)
**R1–R2 Act:** `calculate_cross_section_area`, `calculate_moment_of_inertia`
**Obs 1–2:** geometric properties computed for $d = 40\,\mathrm{mm}$

---

**R3 Act:** `determine_truss_allowable_load(force_direction=inward)`
**Obs 3:** allowable external load $\approx 68.90\,\mathrm{kN}$

---

**R4 Act:** `determine_truss_allowable_load(force_direction=outward)`
**Obs 4:** allowable external load $\approx 194.89\,\mathrm{kN}$

---

**R5–R6 Act:** `visualize_truss_structure` (inward, outward)
**Obs 5–6:** renders truss + member-force annotations for both directions

---

**Final answer:**
Only reports $68.9\,\mathrm{kN}$ **(missing the two-part answer and the direction-change conclusion).**
**Judge:** `is_correct=False`

*Figure 9.* A controlled comparison on the same case: the stronger variant produces a direction-sensitive two-part answer, whereas the weaker variant computes both force directions but outputs an incomplete final response.

**Case 46** (Cell Biology)

**Problem:** You have transformed E. coli cells with pUC18 (pUC origin, AmpR), but want to swap that plasmid with pACYC184 (p15A origin, TetR). Do you expect this to be an easy or difficult manipulation? How would you do this?
**Reference answer:** This would be difficult. pUC18 is a high copy number plasmid and would be hard to remove.

---

**Full interaction trace (claude-sonnet-4-20250514)**
**R1 Act:** `create_plasmid_database()`
**Obs 1: ERROR: database file not found (path redacted)**

---

**R2 Act:** `query_plasmid_properties(plasmid=\"pUC18\")`
**Obs 2:** pUC18: copy_number=500; origin=pMB1; resistance=Ampicillin; size=2.686 kb

---

**R3 Act:** `query_plasmid_properties(plasmid=\"pACYC184\")`
**Obs 3:** pACYC184: copy_number=15; origin=p15A; resistance=Chloramphenicol/Tetracycline; size=4.245 kb

---

**R4 Act:** `calculate_transformation_difficulty(pUC18 -> pACYC184)`
**Obs 4:** difficulty_score=8.54/10; category=Difficult; primary challenge=high-copy plasmid removal

---

**R5 Act:** `analyze_plasmid_replacement_strategy(pUC18 -> pACYC184)`
**Obs 5:** origin compatibility: compatible (no origin conflict); recommended approach: sequential transformation with curing
Recommended protocol (tool output):
1) Prepare competent cells without existing plasmid; 2) Cure/remove the source plasmid (chemical curing or temperature treatment);
3) Verify plasmid loss (PCR or antibiotic sensitivity); 4) Transform the target plasmid using a high-efficiency method;
5) Select on the appropriate antibiotic medium; 6) Confirm transformation by colony PCR

---

**R6 Act:** `visualize_plasmid_comparison(pUC18, pACYC184)`
**Obs 6:** output figure saved to `xxxx.png`

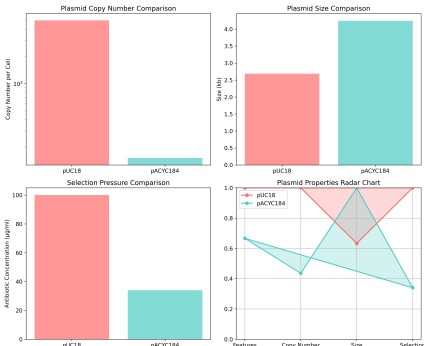

---

**Final answer:**
This is a challenging operation (difficulty score: 8.54/10). The main challenge lies in the high copy number of pUC18 (approximately 500 copies), which must be completely eliminated in order to successfully transition to the low-copy-number plasmid pACYC184 (approximately 15 copies). The recommended procedure includes: (1) curing the source plasmid pUC18 using chemical agents or temperature treatment; (2) verifying plasmid loss by PCR or antibiotic sensitivity testing; (3) introducing pACYC184 using a high-efficiency transformation method; (4) selecting transformants on tetracycline-containing medium; and (5) confirming successful transformation by colony PCR. The entire process typically requires 5–7 days, with an estimated success rate of approximately 60%.
**Judge:** is_correct = True

*Figure 10.* Case 46: Full tool-augmented reasoning trace for a plasmid replacement task, combining database-backed knowledge retrieval and procedural planning.

**Loop Case – qwen3-vl-8b-int-sft, Case 67 (Mechanics)**

**Rounds:** 50 (terminated at tool-round limit; no boxed answer extracted)

**Problem:**

As shown in Figure a, a component consists of a solid steel circular rod with a diameter of $d_1$ and an aluminum circular tube with inner and outer diameters of $d_2$ and $D_2$, respectively. The right end of the component is constrained as a fixed end, and the left end is welded integrally with a rigid circular disc. The length of the component is $l$. Before applying the external load, there is no stress inside the rod and tube. A torsional couple $M_e$ is applied to the rigid circular disc at the left end (as shown in the figure as vector $\boldsymbol{M}_e$). It is given that $d_1 = 50\,\text{mm}$, $d_2 = 60\,\text{mm}$, $D_2 = 76\,\text{mm}$, $l = 500\,\text{mm}$; shear modulus of steel $G_1 = 77\,\text{GPa}$, allowable shear stress $[\tau]_1 = 120\,\text{MPa}$; shear modulus of aluminum $G_2 = 27\,\text{GPa}$, allowable shear stress $[\tau]_2 = 70\,\text{MPa}$. Determine: the maximum torque couple moment that the component can withstand.

**Reference answer:** $6.32\,\text{kN} \cdot \text{m}$

**Input image**

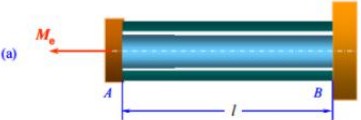

**Registered tools**

`calculate_polar_moment_solid`, `calculate_polar_moment_hollow`, `calculate_torsional_stiffness`, `calculate_max_shear_stress`

**Loop interaction trace**

**R1 Act:** `calculate_polar_moment_solid(diameter=50 mm)`
**Obs 1:** $I_{p,1} = 6.135923 \times 10^5\,\text{mm}^4$

**R2 Act:** `calculate_polar_moment_hollow(outer=76 mm, inner=60 mm)`
**Obs 2:** $I_{p,2} = 2.002979 \times 10^6\,\text{mm}^4$

**R3 Act:** `calculate_torsional_stiffness(G=77 GPa, Ip=Ip1, l=500 mm)`
**Obs 3:** $k_1 = 9.449322 \times 10^7\,\text{N} \cdot \text{mm}$

**R4 Act:** `calculate_torsional_stiffness(G=27 GPa, Ip=Ip2, l=500 mm)`
**Obs 4:** $k_2 = 1.081609 \times 10^8\,\text{N} \cdot \text{mm}$

**R5 Act:** `calculate_max_shear_stress(torque=100000 N*mm, Ip=Ip1, r=25 mm)`
**Obs 5:** steel $\tau = 4.074367\,\text{MPa}$

**R6 Act:** `calculate_max_shear_stress(torque=100000 N*mm, Ip=Ip2, r=38 mm)`
**Obs 6:** aluminum $\tau = 1.897174\,\text{MPa}$

**R7–R50 (degenerate loop): repeatedly re-invokes `calculate_max_shear_stress` with identical inputs, alternating between the same two parameter sets (steel vs. aluminum), without updating the torque or applying the allowable-stress constraints.**

**Termination:**

`boxed_extraction_success=False`. The run ends due to the max tool-round limit.

**Final message:**

**"Tool-round limit exceeded; stopped automatically."**

*Figure 11.* Case 67: Failure under round-limit termination due to repetitive tool calls without progress.

---

**Case 53 (Materials Science)**

**Problem:** From the compound's MS/MS spectrum ¡image¿, determine the number of chlorine atoms based on isotope pattern and other compound information in the spectrum.

**Reference answer:** 1

**Input image:**

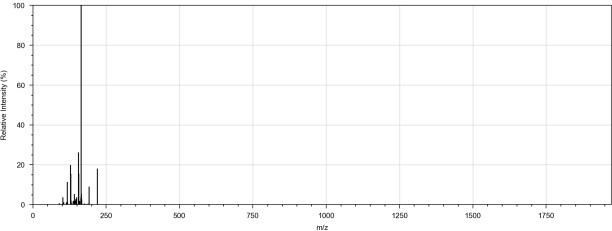

**Phase 1: Environment setup (shared)**

Tool registry: `toolkits.life_science.mass_spectrometry.mass_spectrometry_tools_gym`

Registered tools (5): `extract_peaks_from_spectrum`, `find_isotope_cluster`, `determine_chlorine_number_from_ratio`, `visualize_isotope_pattern_comparison`, `visualize_mass_spectrum`

---

**Phase 2: Per-model interaction trace + evaluation**

(Each model block below includes rounds, key tool-use steps, and the final evaluation verdict.)

**Model:** `gemini-2.5-pro-thinking-2048`   (**Rounds:** 3; **Final:** Wrong)

**R1**: `extract_peaks_from_spectrum` ⇒ peaks `[111,146,148]` (intensity `[27,100,65]`)

**R2**: `find_isotope_cluster` ⇒ ratio(M+2/M)=0.65

**R3**: `determine_chlorine_number_from_ratio` ⇒ num_chlorine=2

**Eval**: boxed=2; standard=1; verdict=**Wrong**

---

**Model:** `glm-4.6v`   (**Rounds:** 50; **Final:** No boxed answer)

**R1–R2**: **tool-call parsing instability** (non-standard Action format; arguments often missing)

**R3–R33**: repeats `extract_peaks_from_spectrum(...)` with identical results (no progress) . . .

**R34–R50**: arguments collapse to `{}`; **repeated validation error: missing required** `mz_values`

**Eval**: boxed answer not found; judge skipped

---

**Model:** `Qwen/Qwen3-VL-235B-A22B-Thinking`   (**Rounds:** 1; **Final:** Correct)

**R1**: `determine_chlorine_number_from_ratio` ⇒ num_chlorine=1

**Eval**: boxed=1; standard=1; verdict=**Correct**

---

**Model:** `gpt-5`   (**Rounds:** 1; **Final:** Correct)

**R1**: `determine_chlorine_number_from_ratio(observed_ratio=0.33)` ⇒ num_chlorine=1

**Eval**: boxed=1; standard=1; verdict=**Correct**

---

**Model:** `qwen3-vl-8b-thinking`   (**Rounds:** 13; **Final:** Correct)

**R1–R11**: repeated `extract_peaks_from_spectrum` with varying thresholds ⇒ no peaks

**R12**: `determine_chlorine_number_from_ratio` ⇒ num_chlorine=1 (medium confidence)

**R13**: visualization call triggers a **TypeError**, then stops

**Eval**: boxed=1; standard=1; verdict=**Correct**

---

**Model:** `claude-sonnet-4-20250514`   (**Rounds:** 6; **Final:** Correct)

**R1–R4**: peak extraction + cluster refinement ⇒ ratio(M+2/M)=0.32

**R5**: `determine_chlorine_number_from_ratio` ⇒ num_chlorine=1

**R6**: visualization call triggers a **TypeError**, then stops

**Eval**: boxed=1; standard=1; verdict=**Correct**

---

*Figure 12.* Case 53: Multi-model tool-use traces and evaluation outcomes on a mass spectrometry reasoning task.

---

### Case 68 (Physics)

**Task:** A homogeneous rod $AB$ with a length of $2b$ and a weight of $P$ is placed on a horizontal surface and a fixed cylinder with a radius of $r$. Assuming the coefficient of friction is $f$ everywhere, find the maximum value of $\phi$ when the rod is in equilibrium.

**Reference answer:** $\sqrt{\frac{fr}{(1+f^2)b}}$

**Phase 1: Environment setup (shared)**

Registered tools (7): `calculate_contact_point_geometry`, `calculate_friction_forces`, `solve_equilibrium_equations`, `verify_analytical_solution`, `plot_rod_configuration`, `plot_equilibrium_analysis`, `plot_parameter_sensitivity`

Input image:

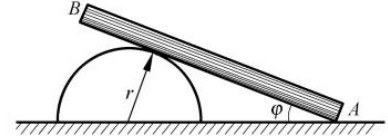

---

**Phase 2: Per-model interaction trace + evaluation**

(Each model block below includes rounds, key tool-use steps, and the final evaluation verdict.)

**Model:** `glm-4.6v` (**Rounds:** 5; **Final:** Correct)
**R1:** `calculate_contact_point_geometry(`$\phi$`=0.5, r=1, b=2)` $\Rightarrow$ geometry computed
**R2:** `solve_equilibrium_equations(...)` $\Rightarrow$ **reports non-physical negative normal forces**
**R3:** `verify_analytical_solution(r=1, b=2, f=0.3)` $\Rightarrow$ closed form verified
**R4–R5:** repeats checks; final response not in Action-format

**Eval:** boxed=$\sqrt{\frac{fr}{(1+f^2)b}}$; verdict=**Correct**

---

**Model:** `Qwen/Qwen3-VL-235B-A22B-Thinking` (**Rounds:** 1; **Final:** Correct)
**R1:** one analytical tool call $\Rightarrow$ numeric check, then stops

**Eval:** boxed=$\sqrt{\frac{fr}{(1+f^2)b}}$; verdict=**Correct**

---

**Model:** `gpt-5` (**Rounds:** 1; **Final:** Correct)
**R1:** one analytical tool call $\Rightarrow$ numeric check, then stops

**Eval:** boxed=$\sqrt{\frac{fr}{(1+f^2)b}}$; verdict=**Correct**

---

**Model:** `gpt-4o` (**Rounds:** 0; **Final:** Correct)
No tool calls; direct final answer

**Eval:** boxed=$\sqrt{\frac{fr}{(1+f^2)b}}$; verdict=**Correct**

---

**Model:** `qwen3-vl-8b-thinking` (**Rounds:** 2; **Final:** Correct)
**R1:** `verify_analytical_solution(...)` $\Rightarrow$ **ERROR: non-numeric parameters**
**R2:** `verify_analytical_solution(...)` $\Rightarrow$ succeeds; closed form returned

**Eval:** boxed=$\sqrt{\frac{fr}{(1+f^2)b}}$; verdict=**Correct**

---

**Model:** `claude-sonnet-4-20250514` (**Rounds:** 5; **Final:** Correct)
**R1–R4:** multiple geometry/equilibrium checks; some intermediate non-physical values reported
**R5:** ends with the correct closed form

**Eval:** boxed=$\sqrt{\frac{fr}{(1+f^2)b}}$; verdict=**Correct**

---

**Model:** `gemini-2.5-pro-thinking-2048` (**Rounds:** 2; **Final:** Wrong)
**R1–R2:** `verify_analytical_solution(...)` repeated; no further refinement

**Eval:** boxed=$\arcsin\left(\sqrt{\frac{fr}{b(1+f^2)}}\right)$; verdict=**Wrong**

---

*Figure 13.* Case 68: Phase-structured multi-model reasoning traces for a classical statics problem with frictional contact.

