# OpenReview forum: "SciAgentGym: Benchmarking Multi-Step Scientific Tool-Use in LLM Agents"
_ICML.cc/2026/Conference — ICML 2026 regular_

### Official Review · Reviewer_9gVW · 2026-03-10

**Soundness:** 3
**Presentation:** 3
**Significance:** 2
**Originality:** 2
**Overall Recommendation:** 5
**Confidence:** 4

**Summary:**

This paper introduces a framework for evaluating scientific reasoning in LLM agents through tool use rather than static knowledge. It presents three contributions: SciAgentGym, an interactive environment equipping agents with domain-specific scientific tools; SciAgentBench, a benchmark spanning multiple scientific domains and difficulty levels; and SciForge, a method for generating logic-aware training data from verified execution trajectories.

**Compliance With Llm Reviewing Policy:**

Affirmed.

**Final Justification:**

Thanks for the authors for addressing my comments. I raised my score.

**Key Questions For Authors:**

None

**Limitations:**

SciForge's generalizability is unclear. Since both the training data and the questions are artificially generated, it is hard to know if these gains would hold on real human-written scientific problems.

**Strengths And Weaknesses:**

Strengths:
1. The paper evaluates a wide range of models across multiple scientific domains and task difficulties, giving a thorough picture of where current models stand.
2. Testing agents with and without tools is a nice design choice that clearly shows how much tools actually help.
3. Training a small 8B model on SciForge's generated data and beating models more than 20 times larger is a strong and convincing result.

Weaknesses:
1. Tool reliability is not discussed. Some tools are known to fail, and the paper never separates how much of the agent's failures are actually the tool's fault rather than the agent's.

---

> ### Author Rebuttal · Authors · 2026-03-29
>
> We sincerely thank the reviewer for the constructive feedback and recognition. We hope the clarifications below help address the concern.
>
> **W1: Tool reliability and fault attribution**
>
> Tool-level errors are expected in our benchmark, as recovery from execution failures is itself a core competency under evaluation (§6.3). The environment deliberately returns fine-grained error diagnostics so that agents can self-correct. On tool reliability and fault attribution, we address this from three perspectives: tool-side quality assurance, systematic fault attribution, and recovery analysis.
>
> **Tool-side quality assurance.** Our three-tier pipeline ensures that tool implementation defects are rare: (1) automated unit testing with a ≥75% pass-rate threshold filters out tools that fail on core tasks; (2) ~593 Category I tools inherit correctness from peer-reviewed libraries (eg. RDKit, PySCF, Biopython); (3) domain PhD students assessed a 10% random sample with a 90% acceptance threshold. The vast majority of errors agents encounter are legitimate signals triggered by agent-side mistakes (e.g., passing "380nm" as a string instead of a numeric value, as in Case 05, Figure 7), not tool bugs.
>
> **Fault attribution.** To directly quantify the split, we performed LLM-based attribution on all 2,244 failed tool-enabled trajectories across 19 models. An LLM judge examined each trace and classified the root cause as: agent-fault (tool worked correctly but agent erred), tool-fault (tool persistently errored despite reasonable agent attempts), mixed, or other.
>
> | Attribution | Count | % |
> |:---|:---:|:---:|
> | agent-fault | 1,661 | 74.0 |
> | tool-fault | 274 | 12.2 |
> | other | 272 | 12.1 |
> | mixed | 37 | 1.6 |
> | tool-related upper bound (tool_fault + mixed) | 311 | 13.9 |
>
> Agent-side failures account for 74.0% of all failures; even attributing all mixed cases to tools, the tool-related upper bound is only 13.9%. By discipline, Physics and Chemistry show the highest agent-fault (\~80%) and lowest tool-fault (\~8%), while Life Science and Materials have higher tool-fault (\~20%), reflecting more complex input formats.
>
> **Recovery separates strong from weak models.** Tool-related errors (\~14%) are not inherently problematic; what matters is whether agents can recover from them. While SR drops from 33.2% to 22.5% (−10.7%) in the presence of tool errors, this gap varies substantially across models:
>
> | Model | SR (w/ tool error) | SR (w/o tool error) | Gap |
> |:---|:---:|:---:|:---:|
> | Grok-4-1 | 40.0 | 37.0 | +3.0 |
> | Claude-Sonnet-4 | 35.8 | 35.6 | +0.2 |
> | GPT-5 | 30.6 | 40.1 | −9.5 |
> | O4-mini | 15.6 | 38.5 | −22.9 |
> | Qwen3-VL-8B-Inst | 10.6 | 26.5 | −15.9 |
> | SciAgent-8B | 25.0 | 27.6 | −2.6 |
>
> Strong models show different resilience to tool errors. Grok-4-1 and Claude-Sonnet-4 are virtually unaffected (+3.0% and +0.2% respectively), while GPT-5 experiences a moderate decline (−9.5%) but still maintains competitive absolute performance (30.6%). In contrast, weak models degrade sharply: O4-mini drops by −22.9% and Qwen3-VL-8B-Inst by −15.9%. This is consistent with the four-dimensional feedback utilization analysis in §6.3 (Figure 4, middle): models with higher Adaptation (responding to error signals), Tuning (correcting parameters), Switching (pivoting to alternative tools), and Loop Escape rates recover effectively, whereas weaker models break down at the earliest stage and cascade into repetitive loops. The key differentiator is not whether tool errors occur, but whether the agent can interpret feedback and adjust accordingly.
>
> **W2: SciForge generalizability**
>
> We would like to clarify a key distinction: SciForge generates training data, while evaluation is always on real, human-authored scientific problems. SciAgentBench's tasks are sourced from five established benchmarks (Table 6: SciInstruct, GPQA, BMMR, SFE, RBench-V), each verified by domain experts. The gains in Table 3 already demonstrate that synthetic training data generalizes to authentic scientific tasks.
>
> **Held-out experiment.** To further validate, we sampled 200 additional problems from the original source benchmarks not included in SciAgentBench, creating a completely independent test set:
>
> | Model | Held-out Accuracy |
> |:---|:---:|
> | Qwen3-VL-8B (Base) | 35.11 |
> | SciAgent-8B | 41.27 |
>
> SciAgent-8B achieves +6.16% on problems never seen during training. Combined with cross-domain transfer evidence in Table 4 (e.g., Chemistry-only training improving Life Science by +7.1% with zero negative transfer), and our detailed case analysis in Appendix F.2 showing that acquired capabilities such as constraint checking, numerical precision, and structured workflow execution transfer as domain-general meta-skills.
>
> Thank you again for your feedback. We have done our best to address your concerns, and we sincerely hope the clarifications are helpful in evaluating our work. We would be grateful if you would consider updating your score in light of these clarifications.

---

> > ### Author Rebuttal · Reviewer_9gVW · 2026-04-02
> >
> > Thanks for the thorough rebuttal. Both concerns are addressed convincingly.
> > On tool reliability, the LLM-based attribution across 2,244 failed trajectories is exactly the kind of systematic analysis this concern required. The 74% agent-fault / 13.9% tool-fault upper bound is clear and well-evidenced. The model-level breakdown showing strong models like Grok-4-1 and Claude-Sonnet-4 being virtually unaffected by tool errors while weaker models degrade sharply is a compelling result that actually enriches the paper's analysis.
> > On SciForge generalizability, the held-out experiment on 200 problems from original source benchmarks not in SciAgentBench directly addresses the concern. A +6.16% gain on unseen human-authored problems is a meaningful and honest test of generalization.
> > Raising my score to 5.

---

> > > ### Author Response · Authors · 2026-04-03
> > >
> > > Thank you so much for your thoughtful follow-up and for taking the time to carefully read our rebuttal. We are truly grateful for your generous and encouraging response, and we greatly appreciate your recognition of the additional analyses on tool reliability and SciForge generalizability. Your updated assessment means a great deal to us. We sincerely thank you again for your time, consideration, and support, and we wish you all the best.
> > >
> > > Dear Reviewer, We noticed that the score currently visible on our side still appears as 4, and were wondering whether the updated rating may not yet have been reflected in the system. If convenient, we would greatly appreciate it if you could kindly check.

---

### Official Review · Reviewer_zNYA · 2026-03-11

**Soundness:** 2
**Presentation:** 3
**Significance:** 3
**Originality:** 3
**Overall Recommendation:** 4
**Confidence:** 2

**Summary:**

This paper introduces a benchmark designed to evaluate large language models on multi-step reasoning and tool usage across multiple scientific disciplines, including physics, chemistry, materials science, and life sciences. The benchmark emphasizes long-horizon tasks that require structured tool interactions, aiming to better reflect real scientific workflows. Based on this benchmark, the authors propose a training method using execution-verified trajectories, which improves model performance on SciAgentBench and leads to noticeable gains over baseline models, particularly on long-horizon tasks.

However, the work also has several limitations. Prior benchmarks such as BioProBench have already explored long-horizon scientific reasoning, which reduces the novelty of the proposed benchmark. The training approach based on failure-augmented trajectories may also benefit from comparison with reinforcement learning–based optimization or workflow-guided methods such as RAG, especially given that many scientific procedures follow relatively deterministic workflows. In addition, the overall accuracy remains relatively low, and performance in some domains (e.g., materials science) is particularly limited, leaving a gap between current results and the goal of building reliable scientific agents.

**Compliance With Llm Reviewing Policy:**

Affirmed.

**Final Justification:**

This paper presents a benchmark for evaluating multi-step reasoning and tool usage across multiple scientific domains, along with a well-motivated training approach using execution-verified trajectories.

The rebuttal effectively addressed my main concerns: it clarified the benchmark’s distinction from prior efforts like BioProBench, provided experimental comparisons showing why supervised trajectory training outperforms RL and RAG-based alternatives, and explained cross-domain transfer limitations, including low performance in the materials science domain.

**Key Questions For Authors:**

1. Clarifying the key differences (e.g., tool interaction, task structure, evaluation design) would help better establish the novelty of the proposed benchmark.
2. Since the proposed method includes failure-based trajectory augmentation, it would be useful to understand whether the authors considered RL-based policy optimization and what advantages their approach provides over RL. Many scientific procedures follow relatively structured workflows. A comparison with retrieval-based baselines could help validate whether the proposed training method provides additional benefits.
3. Additional analysis of cross-domain transferability would help clarify the robustness of the proposed approach (e.g., 9.1 for Qwen3-VL-8B-Merged). Further clarification on how these results support the reliability claim would help better contextualize the experimental findings.

**Limitations:**

yes

**Strengths And Weaknesses:**

**strengths
1. The paper proposes a benchmark designed for scientific domains to evaluate large language models on multi-step reasoning and tool usage across multiple scientific disciplines. The benchmark covers areas such as physics, chemistry, materials science, and life sciences.
2. Based on this benchmark, the authors introduce a training approach based on execution-verified trajectories. Experimental results show that the proposed method improves the model’s success rate on SciAgentBench, achieving noticeable gains in both overall performance and long-horizon tasks compared with the baseline models.

**weaknesses
1. I acknowledge the authors’ contribution in proposing a benchmark that targets cross-disciplinary long-horizon reasoning tasks. However, long-horizon task evaluation has already been explored in some prior works (e.g., BioProBench). This somewhat weakens the distinction between the proposed benchmark and existing efforts. It would be helpful for the authors to further clarify the unique aspects and advantages of their benchmark compared with these prior datasets.
 - BioProBench: Comprehensive Dataset and Benchmark in Biological Protocol Understanding and Reasoning - https://arxiv.org/abs/2505.07889

2. The proposed training approach for long-horizon tasks, particularly the trajectory augmentation based on failure behaviors, raises an interesting question regarding training paradigms. Such mechanisms may potentially be more suitable for reinforcement learning–based policy optimization, which is often considered more appropriate for tasks with uncertain or adaptive long-horizon decision processes. It would be valuable for the authors to discuss the advantages of the proposed supervised trajectory training approach compared with reinforcement learning methods.
In addition, although the benchmark focuses on long-horizon reasoning, many experimental procedures in domains such as physics and chemistry typically follow relatively deterministic workflows. In such cases, it may be feasible to define workflows through retrieval-augmented generation (RAG) or similar knowledge retrieval mechanisms. Therefore, it would strengthen the paper if the authors could provide comparisons with alternative training strategies or workflow-guided approaches to further validate the effectiveness of the proposed method.

3. From the experimental results, the overall accuracy remains relatively low. While this partially demonstrates the challenging nature of the proposed benchmark, there is still a considerable gap between the current performance and the paper’s claim of enabling reliable scientific agents. Furthermore, the performance of Qwen3-VL-8B-Merged in the materials science domain is only 9.1, which appears insufficient to fully support the paper’s claim regarding cross-domain transferability.

---

> ### Author Rebuttal · Authors · 2026-03-29
>
> We thank the reviewer for these insightful questions. We will address three aspects: comparison with existing benchmarks, training method comparisons, and domain transfer evaluation.
>
> **W1 & Q1: Novelty relative to BioProBench and existing benchmarks**
>
> BioProBench measures static language understanding in which models read protocol descriptions and generate text, a setting that RAG naturally supports. SciAgentGym measures interactive scientific execution: agents must plan, invoke tools, recover from failures, and manage state across multi-turn trajectories in a live environment with 1,780 typed tools. RAG cannot replace tool use here, just as reading a lab manual cannot replace running the experiment. This distinction also extends to evaluation: while BioProBench evaluates generated text against references, our benchmark measures agent execution itself, with SR capturing final success and SPL assessing trajectory efficiency against an expert-verified reference path. Beyond final outcomes, SciAgentGym further supports trajectory-level analysis of long-horizon scientific tasks. We will add the BioProBench comparison to Related Work and Table 1.
>
> **W2 & Q2: Comparison with RL-based optimization and RAG-based workflow guidance**
>
> For **RL**, we ran GRPO on Qwen3-VL-8B with format reward (tool-call format compliance) and outcome reward (final-answer matching).
>
> |   |   |   |
> |---|---|---|
> |Setting|SR(%)|Δ|
> |Base|23.4|—|
> |+RL (100/200/300/400 steps)|24.6/26.2/25.1/24.8|+1.2~+2.8|
> |+SFT (SciForge)|30.1|6.7|
>
> RL brings only limited and unstable gains (+2.8 at most), substantially below SFT (+6.7). We attribute this to three factors: (1) sparse rewards and difficult credit assignment, since final-answer correctness is the only quality-sensitive signal; (2) insufficient exploration, as 16 rollouts per group are inadequate for an action space involving 1,780 tools and typically 30–70 candidate tools per task; and (3) multimodal rollout complexity, since tools often generate intermediate images that must be incorporated into later reasoning. These results support our core finding (§6.3) that the key bottleneck is error correction from execution feedback, which is better addressed by SFT with error-recovery trajectories (§5.2) than by RL with sparse outcome rewards.
>
> **RAG Comparison：** We built a RAG baseline with all knowledge cards from the toolkit, using BM25 + vector re-ranking (top 4 injected into the prompt):
>
> |   |   |   |   |
> |---|---|---|---|
> |Model|RAG|w/o Tools|w/ Tools|
> |GPT-5|35.9|32.3|41.3|
> |Claude-Sonnet-4|26.3|22.4|35.9|
> |Gemini-2.5-Flash|28.6|28.5|32.7|
> |Qwen3-VL-235B-Inst|22.4|23|23.9|
> |Qwen3-VL-8B-Inst|17.3|18.4|23.4|
>
> RAG improves over the w/o tools setting but consistently underperforms w/ tools (ReAct). This is expected because scientific tool use requires not only selecting relevant tools, but also parameter binding, type matching, and iterative trial-and-revision across tools, which RAG alone cannot support. For these tasks, interactive tool use with execution feedback remains essential.
>
> **W3 & Q3:**
>
> **On the Claim of "Reliable Scientific Agents".** We respectfully note that the paper does not make this claim. Our conclusion states that the results "underscore the promising potential of next-generation autonomous scientific agents," which is intended as encouragement for continued community attention rather than a claim of solved reliability. The sharp drop from L1 to L3 (46.4% → 14.7%, Table 3) further reflects the benchmark's discriminative power and the substantial headroom that remains.
>
> **On cross-domain transfer evidence.** Our transfer claims are primarily based on single-domain ablations in Table 4. Training on Chemistry alone improves Physics, Materials, and Life Sciences; training on Physics alone improves the other three, without ever seeing those domains' data. Case analysis (Appendix F.2, Figure 9) shows that acquired capabilities (constraint checking, numerical precision, structured workflow execution) transfer as meta-skills. Every single-domain model outperforms the baseline across all four domains with zero negative transfer.
>
> **On the model's materials improvement.** We attribute the Merged model's limited Materials improvement to data imbalance, not negative transfer: (1) Materials has smaller data volume, diluted when merged with larger Physics/Chemistry sets. (2) Materials tasks involve highly specialized representations (POSCAR structures, sparse coordinates); even Gemini-2.5-Pro-Thinking fails (Case 53, Figure 12). (3) Materials is generally a difficult domain for current models, likely due in part to limited relevant pretraining knowledge. Despite this, the Merged model achieves the best overall SR (30.1%), top scores in 3/4 domains, and the largest total gain (+7.0%).
>
> Thank you again for your feedback. We hope we've addressed your concerns and strengthened your confidence in our paper, and we would greatly appreciate it if you could update your score.

---

> > ### Author Rebuttal · Reviewer_zNYA · 2026-04-02
> >
> > The authors’ rebuttal largely addresses my concerns, providing satisfactory clarifications and supporting evidence, and I will raise my score accordingly.

---

> > > ### Author Response · Authors · 2026-04-02
> > >
> > > Thank you so much for your time, thoughtful consideration, and generous follow-up. We truly appreciate the care you took in reading our rebuttal and reconsidering the paper. We are sincerely grateful for your recognition, and we wish you all the best!

---

### Official Review · Reviewer_bDFV · 2026-03-12

**Soundness:** 2
**Presentation:** 3
**Significance:** 2
**Originality:** 3
**Overall Recommendation:** 4
**Confidence:** 3

**Summary:**

This paper addresses the evaluation and improvement of LLM agents' multi-step tool-use capabilities in scientific domains, presenting two complementary contributions: (1) SciAgentGym, a scalable interactive environment comprising 1,780 domain-specific tools across four natural science disciplines (physics, chemistry, materials science, and life sciences); (2) SciAgentBench, an evaluation suite with three difficulty levels (L1/L2/L3), totaling 259 tasks and 1,134 sub-questions. The evaluation reveals a key bottleneck: GPT-5's overall success rate is only 41.3%, dropping sharply from 60.6% to 30.9% as the number of interaction turns increases. To address this, the authors propose SciForge, a data synthesis method that models the tool action space as a dependency graph to generate logic-aware training trajectories; SciAgent-8B, fine-tuned with SciForge, surpasses Qwen3-VL-235B-Instruct (+6.7%) and demonstrates positive cross-domain transfer.

**Compliance With Llm Reviewing Policy:**

Affirmed.

**Final Justification:**

The paper is technically solid and makes a useful contribution to the study of reasoning over real-world files. The authors’ rebuttal clarified several important details and improved the overall clarity of the submission.

**Key Questions For Authors:**

1. What is the construction pipeline for the 1,780 tools? How many tools have been verified by domain experts? What is the conformance rate between tool implementations and corresponding scientific literature?

2. Among the training trajectories generated by SciForge, how many have undergone manual review? What proportion of the 1,780 tools does the dependency graph cover?

3. What are the per-discipline performance breakdowns? How much does SciAgent-8B improve over baselines in physics/chemistry/materials science/life sciences respectively? Can data on specific cross-domain transfer directions (which discipline transfers to which) be provided?

**Limitations:**

yes

**Strengths And Weaknesses:**

### Strengths

The scale of 1,780 tools and 259 tasks represents a significant advantage among scientific agent benchmarks. The three-level difficulty stratification (L1 meta-operations → L2 short sequences → L3 long trajectories) is well-designed, enabling systematic tracking of the capability gradient from basic tool use to long-trajectory planning, while covering four natural science disciplines to achieve genuine cross-domain evaluation.

SciForge's approach of explicitly modeling tool dependencies as graph structures for trajectory generation is a valuable idea: compared to random tool combinations, the dependency graph ensures logical consistency of training trajectories, and stage-aware sampling further improves data efficiency. The result of an 8B small model surpassing a 235B large model is convincing.

The revealed "long-trajectory bottleneck" (success rate halving as horizon increases) is a valuable diagnostic of current SOTA models, pointing the community toward a clear direction for improvement.

### Weaknesses

1. Tool quality and benchmark reliability are questionable. The construction process, validation methods, and distribution of covered scientific sub-domains for the 1,780 domain-specific tools are not described in detail. Tool correctness (whether implementations are consistent with scientific literature) and test case reliability (whether answers are expert-verified) directly affect benchmark credibility, yet the paper devotes little attention to this and lacks systematic benchmark quality assessment.

2. Quality analysis of SciForge-generated training trajectories is insufficient. Whether dependency-graph-generated trajectories contain erroneous steps, the graph's coverage rate (how many of the 1,780 tools are included in the dependency graph), and the proportion of trajectories manually verified are all unspecified. Training data quality directly determines the credibility of SciAgent-8B results, and the absence of this information raises concerns about reproducibility.

---

> ### Author Rebuttal · Authors · 2026-03-29
>
> We sincerely value your constructive comments. A detailed response and supplementary experiments are provided below.
>
> **W1 & Q1: Tool Construction Pipeline and Quality Assurance.**
>
> This is a fair concern; we clarify below. Tool construction follows a 5-stage pipeline (§3, Appendix B): pattern extraction from ~5,000 problems → package encapsulation → taxonomy organization (Table 8) → automated unit testing (≥75% threshold)  → expert verification, covering 27 sub-domains (Figure 3). Tools fall into three categories with distinct confidence levels:
>
> |   |   |   |   |
> |---|---|---|---|
> |Type|[#Tools](#Tools)|Upstream Basis|Confidence|
> |I: Upstream Library Wrappers|593|RDKit, PySCF, ASE, BioPython, PyMatGen, APBS, PubChem API, RCSB PDB,Materials Project, CIR API|≥90%: correctness inherited from peer-reviewed libraries|
> |II: Scientific Workflow Implementations|742|NumPy, SciPy, SymPy|75–90%: verified against analytical solutions; known gaps in open-system and boundary-condition diversity|
> |III: Visualization & Reporting|445|Matplotlib, Seaborn|≥90%: correctness structurally dependent on upstream Type I/II data|
>
> The key design principle is that Type I tools (593) do not reimplement scientific algorithms but delegate core computation to de facto standard packages; Type II tools are grounded in classical equations with step-wise verifiability against analytical ground truth; Type III tools are faithful renderers of upstream verified data.
>
> Quality control: All tools are tested against ground truth derived from first principles before writing assertions, rather than from observed outputs, with ≥3 test cases per tool covering normal, boundary, and exception conditions. The ≥75% threshold (rather than 100%) intentionally retains tools that are correct in core computation but exhibit undefined behavior at physically unrealizable boundary conditions. To our knowledge, this is more rigorous than existing benchmarks such as ToolBench and API-Bank, neither of which applies systematic unit testing with scientifically grounded assertions.
>
> **Expert verification.** A stratified 10% random sample (200 tools) was reviewed by domain-specific PhD students on three criteria: implementation correctness vs. scientific literature, type safety, and edge case handling. The acceptance rate exceeded 90%.
>
> **W2 & Q2: SciForge Trajectory Quality and Dependency Graph Coverage.**
>
> The dependency graph G_d=(V_d,E_d) uses the full tool set V_d as nodes (Section 5.1, Eq. 2), so node-level coverage is 100%. We believe you may be asking about coverage in the final training data. In the final dataset, 1,365 tools appear at least once, covering 76.7% of the 1,780 tools.
>
> As described in Section 5.2, SciForge first samples executable program graphs from the dependency graph and then executes them in SciAgentGym. Successful executions are retained as Golden Traces (Eq. 5), while failed executions are included only in an explicit error-recovery format and paired with corrected re-execution (Eq. 6), rather than being silently mixed into clean trajectories as unlabeled noise. This yields an automatic retention rate of 97.2% (the filtered portion consisting mainly of incomplete trajectories), with the final dataset comprising 76.4% Golden Traces and 23.6% error-recovery trajectories.
>
> To further assess data quality, we manually audited 200 randomly sampled trajectories from the 11,074 used for SFT across four dimensions:
>
> |   |   |
> |---|---|
> |Dimension|Score (/5)|
> |Tool output correctness|4.09|
> |Parameter format compliance|4.29|
> |Execution integrity|4.94|
> |Answer completeness|4.47|
> |Overall average|4.47|
>
> Execution integrity scored highest (4.94/5), confirming structural completeness of Golden Trace and error-recovery chains. Since the training data are generated at scale, we place particular emphasis on trajectory completeness and executability, and for this reason, all such trajectories are retained for SFT.
>
> **Q3: Per-Discipline Breakdowns and Cross-Domain Transfer.**
>
> This information is already reported in our submission. Table 3 (page 6) provides per-discipline SR for all 19 models; Appendix Table 5 (page 12) gives the full with-tools/without-tools breakdown. SciAgent-8B improvements: Physics +9.0%, Chemistry +6.6%, Life Sciences +6.9%, Materials +2.0%.
>
> Cross-domain transfer is systematically ablated in Table 4. Key findings (Section 6.4): (1) Positive cross-domain transfer within science: chemistry-only training improves Life Science by +7.1% and Materials by +12.9%, suggesting transferable meta-skills (constraint checking, numerical precision). (2) Multi-domain training is best overall (+7.0%). (3) Non-scientific tool data causes negative transfer (−4.6%), confirming gains stem from shared scientific reasoning paradigms, not data volume.
>
> We are grateful for the time you invested in review. We have carefully addressed each of your concerns, and we hope the clarifications and supplementary analyses above are helpful in evaluating our work.

---

> > ### Author Rebuttal · Reviewer_bDFV · 2026-04-03
> >
> > The authors’ rebuttal has addressed most of my main concerns by clarifying the tool construction and quality-control pipeline, as well as providing additional evidence on SciForge trajectory quality and coverage. Based on these clarifications, I consider my concerns to be largely resolved, while the remaining issues are mostly about further strengthening the paper rather than undermining its core contributions.

---

> > > ### Author Response · Authors · 2026-04-04
> > >
> > > We sincerely thank the reviewer for the positive acknowledgement and thoughtful comments. We are very glad that our rebuttal and additional analyses helped address your main concerns. We also truly appreciate your constructive suggestions, which helped us clarify and strengthen the paper.

---

### Official Review · Reviewer_qWU6 · 2026-03-13

**Soundness:** 3
**Presentation:** 3
**Significance:** 3
**Originality:** 3
**Overall Recommendation:** 4
**Confidence:** 3

**Summary:**

This paper introduces SciAgentGym, an integrated platform for evaluating and training LLM agents on multi-step scientific tool-use tasks. The contribution is threefold: (1) an interactive environment with 1,780 domain-specific scientific tools across physics, chemistry, biology, and materials science, featuring type-safe I/O signatures and sandboxed execution; (2) SciAgentBench, a hierarchical benchmark of 259 tasks (1,134 sub-questions) at three difficulty levels (L1–L3), curated from five existing scientific benchmarks and validated by domain experts; and (3) SciForge, a data synthesis method that leverages tool dependency graphs to generate logically coherent training trajectories, including error-recovery augmentation. Experiments show that all models exhibit sharp performance degradation with increasing task complexity (GPT-5: 60.6% L1 → 30.9% L3), and that SciAgent-8B fine-tuned on SciForge-generated data surpasses models 30× its size (Qwen3-VL-235B-Instruct). Detailed error recovery analysis identifies adaptation failure (32.9%) and parameter tuning failure (6.6%) as core bottlenecks.

**Compliance With Llm Reviewing Policy:**

Affirmed.

**Key Questions For Authors:**

1. **Answer matching criteria:** How exactly are sub-question answers compared to ground truth? Is it exact string match, numerical match within tolerance, or LLM-based semantic comparison? For scientific computations, what tolerance thresholds are used, and how sensitive are the results to these thresholds?

2. **Why not RL?** Given that the environment provides step-by-step execution feedback and the error recovery analysis highlights feedback utilization as the core bottleneck, have you experimented with RL-based training (e.g., using execution success/failure as rewards)? If so, what were the results? If not, what is the rationale for limiting to SFT?

3. **Tool exposure during evaluation:** How many tools are exposed to the agent per task on average? Is the agent given the full set of 1,780 tools, a domain-relevant subset, or a task-specific subset? This significantly affects the difficulty of tool selection and the comparability of results.

4. **Intermediate step correctness:** Have you analyzed cases where the final answer is correct but intermediate steps contain errors (compensating errors)? How prevalent is this, and does it affect the validity of the SR metric?

5. **Cross-domain transfer:** SciAgent-8B is trained on SciForge data across all four domains. Have you evaluated domain-specific fine-tuning vs. multi-domain fine-tuning? Does performance on one domain benefit or hurt performance on others?

**Limitations:**

The authors adequately discuss the main limitations, including the performance ceiling of current models and the difficulty of error recovery. However, some additional limitations deserve mention: (1) the benchmark is limited to four natural science domains—generalization to engineering, computational science, or interdisciplinary problems is untested; (2) the sandbox environment, while providing safe execution, may not capture all real-world scientific workflow complexities (e.g., long-running simulations, hardware-dependent computations); (3) the reliance on existing benchmarks for task sourcing means the difficulty distribution and domain coverage are inherited rather than deliberately designed. Societal impact discussion is adequate.

**Strengths And Weaknesses:**

**Strengths:**

**Soundness:**
- The benchmark design is rigorous: tasks are filtered through a four-stage pipeline (aggregation → difficulty filtering → execution verification → expert validation), ensuring tasks require genuine multi-step scientific reasoning.
- The evaluation metric (Success Rate) uses strict all-or-nothing scoring—all sub-questions must be correct—which prevents inflated performance from partial credit. SPL further penalizes redundant tool calls using expert-verified shortest paths.
- The error recovery analysis is a standout contribution: decomposing recovery into Adaptation, Tuning, Switching, and Loop Escape provides actionable insights into agent failure modes, going beyond simple accuracy reporting.
- The SciForge data synthesis method is well-designed: sampling executable program graphs from tool dependency structures ensures training trajectories are logically coherent, and the error-recovery augmentation adds realistic failure-correction patterns.

**Presentation:**
- The paper is well-structured with clear motivation, comprehensive experimental results, and insightful qualitative analysis (e.g., Case 05 DTypePromotionError, Case 68 near-miss mathematical error).
- The tool environment architecture is thoroughly described, including type safety mechanisms, sandbox isolation, and the distinction between atomic primitives and composite operations.
- The difficulty stratification (L1/L2/L3) with clear definitions makes results interpretable and enables fine-grained analysis of model capabilities.

**Significance:**
- Addresses a genuine gap: existing agent benchmarks focus on general-purpose tool use, while scientific workflows require domain-specific tools with complex type systems and dependency structures.
- The finding that small fine-tuned models (8B) can outperform much larger models (235B) on scientific tool use is practically important and demonstrates the value of domain-specific training data.
- The error recovery analysis reveals that error signal utilization—not tool knowledge—is the primary bottleneck, which has implications for future agent training research.

**Originality:**
- The scientific tool type system (SMILES, Vector3D, ProteinStructure, POSCAR, etc.) and tool dependency graph formalization are novel contributions that distinguish this from general-purpose agent benchmarks.
- SciForge's approach of sampling from tool dependency graphs to generate training data is a creative solution to the scarcity of scientific agent trajectories.
- The multi-dimensional error recovery taxonomy (Adaptation/Tuning/Switching/Loop Escape) provides a new analytical framework for understanding agent failures.

---

**Weaknesses:**

**Soundness:**
- **Sub-question evaluation criteria underspecified (Moderate):** The paper does not detail how sub-question answers are compared against ground truth—exact match, numerical tolerance, or semantic matching? For scientific computations where floating-point precision matters, this is a critical detail that affects reproducibility and fairness of comparisons.
- **Only final answers evaluated (Moderate):** While the all-or-nothing SR metric is strict, intermediate reasoning steps are not independently verified. A model could reach correct final answers through incorrect intermediate reasoning (compensating errors), which the current evaluation would not catch.
- **Limited training paradigm exploration:** SciForge uses only SFT. Given the sequential decision-making nature of tool use, RL-based approaches (e.g., with environment rewards) could potentially yield better error recovery. The paper does not discuss why RL was not explored.

**Presentation:**
- The paper could benefit from a clearer discussion of how task-relevant tools are selected and exposed to agents during evaluation (briefly mentioned in §E.1 but important for reproducibility).
- Some key statistics (e.g., average number of tools per task, average trajectory length) are missing from the main text.

**Significance:**
- Tasks are sourced from five existing benchmarks rather than being originally designed, which somewhat limits the novelty of the benchmark itself (though the curation and integration are valuable).
- The four scientific domains are all natural sciences; engineering, social science, and computational domains are not represented.

**Originality:**
- The overall paradigm of "interactive environment + benchmark + training data synthesis" follows established patterns in agent research (e.g., WebArena, SWE-bench), adapted to the scientific domain. While the adaptation is non-trivial, the conceptual framework is not entirely new.

---

> ### Author Rebuttal · Authors · 2026-03-29
>
> We greatly appreciate the reviewer's thoughtful comments. Below, we offer detailed responses and supplementary experiments.
>
> **Q1.** Our evaluation uses a hierarchical matching strategy (Appendix E.1): (1) strict recursive JSON matching with numerical tolerance 0.05; (2) LLM-based semantic verification (GPT-4.1) when strict matching fails only on text fields (Tables 13–14 provide the complete prompts); (3) binary all-or-nothing scoring. We tested tolerance sensitivity at 0.01/0.05/0.10:
> |   |   |   |   |
> |---|---|---|---|
> |Model|0.01|0.05|0.1|
> |GPT-5|41.2|41.3|41.7|
> |Claude-Sonnet-4|35.5|35.9|36.3|
> |Qwen3-VL-235B|23.5|23.9|24.2|
> |Qwen3-VL-8B|23.1|23.4|23.8|
> We observe that (1) tolerance has minimal impact on results. Relaxing the threshold from 0.01 to 0.10 changes SR by at most 1.5 points across all models, suggesting that correct answers already satisfy the strictest threshold, while incorrect answers remain too far off to pass under any threshold. (2) model rankings and main conclusions remain unchanged. These results support 0.05 as a reasonable default tolerance.
>
> **Q2.** We did conduct RL experiments using GRPO on Qwen3-VL-8B, with a format reward (tool-call compliance) and an outcome reward (final-answer correctness only):
> |   |   |   |
> |---|---|---|
> |Setting|SR(%)|Δ|
> |Base|23.4|—|
> |+RL (100/200/300/400 steps)|24.6/26.2/25.1/24.8|+1.2~+2.8|
> |+SFT (SciForge)|30.1|6.7|
>
> RL yields limited and unstable gains (+2.8 at most), compared with +6.7 from SFT. We attribute this to three agentic RL challenges: (1) sparse rewards and hard credit assignment: final-answer correctness is the sole quality-sensitive signal, with format rewards blind to tool call quality, compounded by long trajectories; (2) limited exploration: 16 rollouts per group is insufficient over an action space of 30–70 candidate tools per task; (3) multimodal complexity: tools frequently yield intermediate image outputs that must be folded into subsequent reasoning.
>
> This reinforces our central finding (§6.3) that error correction from execution feedback is the dominant bottleneck. Sparse outcome rewards do not provide step-level signals for localizing failures, whereas SFT on error-recovery trajectories (§5.2) directly teaches the agent how to recover and therefore yields stronger performance at this stage. We may leave process reward models to future work.
>
> **Q3.**: Agents are evaluated with subdomain-level tool sets, averaging about 56 tools per task and ranging from 9 in Protein Structure Analysis to 248 in Atomic & Molecular Physics, rather than with all 1,780 tools or a minimal subset. This design balances realism and tractability: scientists are typically familiar with their subdomain, yet still need to choose among many tools. It also ensures fair cross-model comparison while preserving meaningful tool-selection difficulty.
>
> **Q4.** SR is an outcome metric, not a process-faithfulness metric, but it is appropriate for SciAgentBench, which evaluates task completion in a closed-loop environment. We audited successful with-tools trajectories for erroneous intermediate steps, including failed tool calls, parameter mistakes, and exploratory actions irrelevant to the final answer.
> |   |   |
> |---|---|
> |Model|% with ≥1 intermediate error|
> |GPT-5|38.20%|
> |Claude-Sonnet-4|45.70%|
> |Qwen3-VL-235B|51.30%|
> |SciAgent-8B|48.90%|
>
> We find that these cases  are better characterized as exploration followed by recovery, rather than classical compensating errors. Agents interact with tools, receive execution feedback, and revise their plans; intermediate failures are natural. The key capability is whether the agent can recover and still complete the task, as analyzed in Section 6.3 (Figure 4). We include metrics that directly target intermediate-step correctness: Tuning (6.6%) measures correct parameter specification, Switching (15.3%) measures tool switching after failure, and Adaptation (32.9%) captures responsiveness to execution feedback. Thus, intermediate-step quality is not assessed by SR alone. We therefore view SR as task completion, not process faithfulness. To complement it, we also report SPL, which penalizes overly long successful trajectories and partially reflects efficiency and recovery cost.
>
> **Q5.** We have already conducted this experiment, with results reported in Table 4 (§6.4, "Ablation: Domain Transfer"). Key findings: (1) Positive cross-domain transfer within science：chemistry-only training improves Life Science by +7.1% and Materials by +12.9%, suggesting transferable meta-skills (constraint checking, numerical precision). (2) Multi-domain training is best overall. (3) Non-scientific tool data causes negative transfer (−4.6%), confirming the gains stem from shared scientific reasoning paradigms, not data volume.
>
> Thank you again for your feedback. We hope the above clarifications sufficiently address your questions and concerns.

---

> > ### Author Rebuttal · Reviewer_qWU6 · 2026-04-03
> >
> > Thanks for the rebuttal, great work

---

> > > ### Author Response · Authors · 2026-04-04
> > >
> > > We sincerely thank the reviewer for the encouraging feedback and positive acknowledgment. We are very glad that our rebuttal and supplementary analyses helped clarify the concerns.

---

### Decision · Program_Chairs · 2026-04-30

**Decision:**

Accept (regular)

**Comment:**

The authors provided three things: (i) SciAgentGym: an interactive environment with ~2K domain-specific tools. (ii) SciAgentBench: a multi-level benchmark to evaluate LLMs on complex, multi-step scientific reasoning tasks. (iii) SciForge: a method for synthesizing logic-aware training trajectories. In general, all reviewers agreed on the robustness of benchmark construction process as well as the strict all-or-nothing evaluation metric which successfully addresses a significant gap in evaluating agentic capabilities within specialized scientific domains. The authors discovered that smaller, fine-tuned models can outperform much larger generalist models coupled with the detailed taxonomy of error recovery failures. This is a particularly valuable contribution.

During the rebuttal, the authors effectively resolved reviewers' concerns about tool reliability, data quality and RL related comments by providing some fault-attribution analyses, cross-language and cross-domain ablation studies and empirical results showing the limitations of RL for certain problem setting. Some reviewers initially raised some concerns on the novelty compared with existing datasets. During rebuttal, the authors successfully clarified that SciAgentGym is focused on interactive and executable tool-use which is distinguished from others. Overall, the authors successfully address the remaining concerns during the rebuttal periods and there is no more primary concerns on this paper. In general, all reviewers satisfied with the paper's technical depth and practical impact. I recommend accepting this paper.